# HOW DO SKIP CONNECTIONS AFFECT GRAPH CONVOLUTIONAL NETWORKS WITH GRAPH SAMPLING? A THEORETICAL ANALYSIS ON GENERALIZATION

## ABSTRACT

Skip connections enable deep Graph Convolutional Networks (GCNs) to overcome over-smoothing, while graph sampling reduces computational demands by selecting a submatrix of the graph adjacency matrix during neighborhood aggregation. Learning deep GCNs with graph sampling has shown empirical success across various applications, but a theoretical understanding of the generalization guarantees remains limited, with existing analyses ignoring either graph sampling or skip connections. This paper presents the first generalization analysis of GCNs with skip connections using graph sampling. Our analysis demonstrates that the generalization accuracy of the learned model closely approximates the highest achievable accuracy within a broad class of target functions dependent on the proposed sparse effective adjacency matrix, denoted by $A^*$. Thus, graph sampling maintains generalization performance when $A^*$ accurately models data correlations. Notably, our findings reveal that skip connections lead to different sampling requirements across layers. In a two-hidden-layer GCN, the generalization is more affected by the sampled matrix deviations from $A^*$ of the first layer than the second layer. To the best of our knowledge, this marks the first theoretical characterization of skip connections' role in sampling requirements. We validate our theoretical results on deep GCNs for benchmark datasets.

## 1 INTRODUCTION

Graph neural networks (GNNs) outperform traditional machine learning techniques when learning graph-structured data, that comprises a collection of features linked with nodes and a graph representing the correlation of the features. As one of the most popular variants of GNN, Graph Convolutional Networks (GCNs) (Kipf & Welling, 2017) perform the convolution operations on graphs by aggregating neighboring nodes to update the feature presentation of every node. GCNs have demonstrated great empirical success such as text classification (Norcliffe-Brown et al., 2018; Zhang et al., 2018) and recommendation systems (Wu et al., 2019; Ying et al., 2018).

As the depth of vanilla GCNs increases, there is a tendency for the node representations to converge toward a common value, a phenomenon known as "over-smoothing" (Li et al., 2018). A widely adopted mitigation approach is to incorporate skip-connections, which allow features to bypass intermediate layers and directly contribute to future layers' output (Li et al., 2019; Xu et al., 2018b). Moreover, skip connections are shown to accelerate the training process (Xu et al., 2021). Skip connections have thus become an essential component of deep GCNs.

Processing large-scale graphs can be computationally demanding, particularly when dealing with the recursive neighborhood integration in deep GCNs. To alleviate this computational burden, graph sampling or sparsification methods select a subset of nodes or edges from the original graph when computing neighborhood aggregation. Various graph sampling approaches have been developed, including node sampling methods like GraphSAGE (Hamilton et al., 2017), layer-wise sampling like FastGCN (Chen et al., 2018), subgraph sampling methods like Graphsaint (Zeng et al., 2020), and other graph sparsification methods like (Li et al., 2020; Chen et al., 2021; You et al., 2020).

While GCNs have demonstrated empirical success, their theoretical foundation remains relatively underdeveloped, particularly the generalization analysis for training GCNs with graph sampling. Du et al. (2019a) provides a theoretical framework connecting graph neural networks and graph kernels. Verma & Zhang (2019); Zhang et al. (2020a); Liao et al. (2020); Maskey et al. (2022);

Tang & Liu (2023) offer convergence and generalization guarantees for GCN without sampling. Xu et al. (2021) proves the implicit acceleration achieved by skip connections in linearized graph neural networks, ignoring graph sampling and non-linear activation. Li et al. (2022a); Zhang et al. (2023b) provide theoretical generalization analyses of training GCNs with sampling. However, their network architectures exclude skip connections and thus are oversimplified for deep GCNs.

To the best of our knowledge, this paper provides the first theoretical generalization analysis of training GCNs with skip connections using graph sampling. We consider the semi-supervised node regression problem, where given all node features and partial labels, the objective is to predict the unknown node labels. Our major results include:

(1) We analyze training two-hidden-layer GCNs with a skip connection, using graph topology sampling methods that prefer edges between low-degree nodes. Our analysis demonstrates that the generalization accuracy of the learned model approximates the highest achievable accuracy within a broad class of target functions, which map input features to labels. Each target function is a sum of a simpler base function that contributes significantly to the output and a more complicated composite function that has a comparatively smaller impact on the output. This class encompasses a wide range of functions, including two-hidden-layer GCNs with skip connections.

(2) This paper proposes the concept of the sparse effective adjacency matrix, denoted as $A^*$, to characterize the influence of graph sampling in GCNs with skip connections. This is an extension from the dense effective adjacency matrix introduced in (Li et al., 2022a) for GCNs lacking skip connections, This work demonstrates that the target functions depend on $A^*$ rather than the original graph's adjacency matrix $A$. $A^*$ can be sparser than $A$. Consequently, even when the sampled adjacency matrix is very sparse and significantly deviates from $A$, as long as it approximates $A^*$ closely enough to enable accurate feature-to-output mapping by the target functions, graph sampling does not compromise the model's generalization performance.

(3) This paper theoretically demonstrates that, owing to the presence of skip connections, sampling in different layers has varying impacts on the model's output. Specifically, the first layer connects directly to the output through the skip connection, and as a result, the deviation of the sampled matrix from the sparse effective matrix $A^*$ has a more significant effect than the sampling deviation of the second layer. In contrast, the second layer influences the output through a composite function that contributes less significantly, allowing for more substantial deviations from $A^*$ in the sampling process without compromising error rates. To the best of our knowledge, this is the first theoretical characterization of how skip connections influence sampling requirements, while previous analyses such as (Li et al., 2022a) assumes that the sampling approach remains consistent across different layers.

## 1.1 RELATED WORKS

**Other theoretical analysis of GNNs** focus on expressive power and convergence analysis. Xu et al. (2018a); Morris et al. (2019) show the power of 1-hop message passing is upper bounded by 1-WL test. Feng et al. (2022); Wang & Zhang (2022) extend the analysis to k-hop message passing neural networks and spectral GNNs. Zhang et al. (2023a) explores the expressive power of GNNs from the perspective of graph biconnectivity. Oono & Suzuki (2020); Ramezani et al. (2020); Cong et al. (2021) investigates the optimization of GNN training.

**Generalization analyses of Neural Networks (NNs).** Various approaches have been developed to analyze the generalization of feedforward NNs. The neural tangent kennel (NTK) approach shows that overparameterized networks can be approximated by kernel methods in the limiting case (Jacot et al., 2018; Du et al., 2019b). The model estimation approach assumes the existence of a ground-truth one-hidden-layer model with desirable generalization and estimates the model parameters using the training data (Zhong et al., 2017; Zhang et al., 2020b; Li et al., 2022b). The feature learning approach analyzes how a shallow NN learns important features during training and thus achieves desirable generalization (Li & Liang, 2018; Allen-Zhu & Li, 2022; 2023). All works ignore the skip connection except Allen-Zhu & Li (2019), which analyzes the generalization of two-hidden-layer ResNet. Our analysis builds upon Allen-Zhu & Li (2019) and extends to GCNs with graph sampling.

**Various GNN sampling methods**. Node-wise sampling (Hamilton et al., 2017) randomly selects nodes and their multi-hop neighbors to create a localized subgraph. Layer-wise sampling (Chen et al., 2018; Zou et al., 2019; Huang et al., 2018) sample a fixed number of nodes in each layer. Subgraph-based sampling (Zeng et al., 2020; Chiang et al., 2019) generates subgraphs by sampling nodes and edges. As for graph sparsification, SGCN (Li et al., 2020) introduces the alternating direction method of multipliers (ADMM) to sparsify the adjacency matrix. UGS (Chen et al., 2021), Early-Bird (You et al., 2020) and ICPG (Sui et al., 2022) design a sparsifying strategy to prune the graph based on the lottery ticket hypothesis.

## 2 TRAINING GCNS WITH LAYER-WISE GRAPH SAMPLING: SUMMARY OF MAIN COMPONENTS

### 2.1 GCN LEARNING SETUP

Let $\mathcal{G} = \{\mathcal{V}, \mathcal{E}\}$ represent an undirected graph, where $\mathcal{V}$ is the set of nodes with $|\mathcal{V}| = N$ nodes and $\mathcal{E}$ is the set of edges. $\Delta$ is the maximum degree of $\mathcal{G}$. An adjacency matrix $\tilde{A} \in R^{N \times N}$ is defined to describe the overall graph topology where $\tilde{A}(i,j) = 1$ if $(v_i, v_j) \in \mathcal{E}$ else $\tilde{A}(i,j) = 0$. $A$ denotes the normalized adjacency matrix with $A = D^{-\frac{1}{2}}(\tilde{A} + I)D^{-\frac{1}{2}}$ where $D$ is the degree matrix with diagonal elements $D_{i,i} = \sum_j \tilde{A}(i,j)$. Each element $A_{ij}$ of the matrix $A$ represents the normalized weight of the edge between nodes $i$ and $j$. $a_n$ denotes the $n$th column of the $A$. Let $X \in \mathbb{R}^{d \times N}$ denote the feature matrix of $N$ nodes, where $\tilde{x}_n \in \mathbb{R}^d$ denotes the feature of node $n$. Assume $\|\tilde{x}_n\| = 1$ for all $n$ without loss of generality. $y_n \in \mathbb{R}^k$ represents the label of node $n$. Let $\Omega \subset \mathcal{V}$ denote the set of labeled nodes. Given $X$ and partial labels in $\Omega$, the objective of semi-supervised node-regression is to predict the unknown labels in $\mathcal{V}/\Omega$.

We consider training a two-hidden-layer GCN with a single skip connection, where the function $\text{out} : \mathbb{R}^{d \times N} \times \mathbb{R}^{N \times N} \to \mathbb{R}^{k \times N}$ with

$$\text{out}(X, A; W, U) = C\sigma(WXA) + C\sigma(U\sigma(WXA)A) \tag{1}$$

where $\sigma(\cdot)$ applies the ReLU activation $\text{ReLU}(x) = \max(x, 0)$ to each entry, $W \in \mathbb{R}^{m \times d}$, $U \in \mathbb{R}^{m \times m}$, and $C \in \mathbb{R}^{k \times m}$ denote the first hidden-layer, second hidden-layer, and output layer weights, respectively. We only train $W$ and $U$. The output of the $n$th node can be written as $\text{out}_n : \mathbb{R}^{d \times N} \times \mathbb{R}^N \to \mathbb{R}^k$ is

$$\text{out}_n(X, A; W, U) = C\sigma(WXa_n) + C\sigma(U\sigma(WXA)a_n) \tag{2}$$

We focus on the $\ell_2$ regression task and the prediction loss of the $n$th node is defined as

$$\text{Obj}_n(X, A, y_n; W, U) = \frac{1}{2}\|y_n - \text{out}_n(X, A; W, U)\|_2^2 \tag{3}$$

The learning problem solves the following empirical risk minimization problem:

$$\min_{W,U} L_\Omega(W, U) = \frac{1}{|\Omega|} \sum_{n \in \Omega} \text{Obj}_n(X, A, y_n; W, U) \tag{4}$$

### 2.2 TRAINING WITH STOCHASTIC GRADIENT DESCENT AND GRAPH SAMPLING

(4) is solved by the stochastic gradient descent (SGD) method starting from random initialization. In each iteration, the gradient of the prediction loss of one randomly selected node is used to approximate the gradient of $L_\Omega$. The recursive neighborhood aggregation through multiplying the feature matrix with $A$ is costly in both computation and memory. Graph sampling sparsify the adjacency matrix $A$ to reduce the computation and memory requirement. For example, edge sampling prunes a subset of edges in $\mathcal{E}$ and the corresponding entries in $A$. One common theme of various edge sampling methods is to keep the edge between nodes with smaller degrees while pruning edges between nodes with larger degrees (Chen et al., 2018; Zeng et al., 2020). The intuition is that if two nodes with smaller degrees are connected, they are more influential to each other, and therefore, the edge between them should be sampled with a higher probability.

In this paper, we allow the sampling methods with different parameter settings in different layers and will characterize how sampling in different layers affects generalization differently. In the $t$th SGD iteration, let $A^{1t}$ and $A^{2t}$ denote the sampled $A$ matrix in the first and second hidden layers, respectively. Let $W^{(t)}$ and $U^{(t)}$ denote the current estimation of $W$ and $U$. When computing the stochastic gradient, instead of (2), we use[1]

$$\text{out}(X, A^{1t}, A^{2t}; W^{(t)}, U^{(t)}) = C\sigma(W^{(t)}XA^{1t}) + C\sigma(U^{(t)}\sigma(W^{(t)}XA^{1t})A^{2t}) \tag{5}$$

---

[1]If different layers use different adjacency matrices, we specify both matrices in the function representation. Otherwise, we use one matrix to simplify notations.

# 3 Main Algorithm and Theoretical Results

## 3.1 Informal Key Theoretical Findings

We first summarize our major theoretical insights and takeaways before formally presenting them.

1. **The first theoretical generalization guarantee of two-hidden-layer GCNs with skip-connection.** We demonstrate that training a single skip-connection two-hidden-layer GCN using our Algorithm 1 returns a model that achieves the label prediction performance almost the same as the best prediction performance using a large class of target functions. We also characterize quantitatively the required number of labeled nodes, referred to as the sample complexity, to achieve the desirable prediction error. To the best of our knowledge, only Li et al. (2022a); Zhang et al. (2023b) provide explicit sample complexity bounds for node classification using graph neural networks, but for shallow GCNs with no skip connection. Our work is the first one that provides a theoretical generalization and sample complexity analysis for the practical GCN architecture with skip connections.

2. **Graph sampling affects generalization through the sparse effective adjacency matrix $A^*$.** We show that training with edge sampling produces a model with the same prediction accuracy as a model trained on a GCN with $A^*$ as the normalized adjacency matrix, i.e., replacing $A$ with $A^*$ in (1). The effective matrix is first discussed in (Li et al., 2022a), in the setup of node sampling for two-hidden-layer GCNs with no skip connection, but $A^*$ in (Li et al., 2022a) is dense. We show that the effective matrix also exists for edge sampling on GCNs with skip connection and can be sparse, indicating that the sampled matrices can be very sparse without sacrificing generalization.

3. **Layer-Specific graph sampling due to skip connection**. We show that in the two-hidden-layer GCN with a single skip connection, the first hidden-layer learns a simpler base function that contributes more to the output, while the second hidden-layer learns a more complicated function that contributes less to the output. Therefore, compared with the first hidden-layer, the second hidden-layer is more robust to graph sampling and can tolerate a deviation of the sampled matrix to $A^*$ without affecting the prediction accuracy. To the best of our knowledge, this is the first theoretical characterization of how skip connections affect the sampling requirements in different layers, while the previous analysis in (Li et al., 2022a) assumes the same matrix sampling deviations for all layers.

## 3.2 Graph Topology Sampling Strategy

Our sampling strategy differs slightly from existing edge sampling methods due to our adjustments aiming to facilitate and simplify the theoretical analysis. Nevertheless, our core concept is still sampling edges between lower-degree nodes with a higher probability, while sampling edges between higher-degree nodes with a smaller probability. This approach aligns with practical edge sampling strategies such as Zeng et al. (2020).

We follow the same assumption on node degrees as that in Li et al. (2022a). Specifically, the node degrees in $\mathcal{G}$ can be divided into $L$ ($L \geq 1$) groups, with each group having $N_l$ nodes ($l \in [L]$). The degrees of all $N_l$ nodes in group $l$ are in the order of $d_l$, i.e., between $cd_l$ and $Cd_l$ for some constants $c \leq C$. $d_l$ is order-wise smaller than $d_{l+1}$, i.e., $d_l = o(d_{l+1})$.

Let matrix $A_{B_{ij}} \in \mathbb{R}^{N_i \times N_j}$ denote a submatrix of $A$ with rows in group $i$ and columns corresponding to group $j$. Then all entries in $A_{B_{ij}}$ are in the order of $\frac{1}{\sqrt{d_i d_j}}$. Note that a relatively smaller entry in $A$ corresponds to an edge between relatively higher-degree nodes. Let $p_{ij}^k$ in $[0, 1]$ ($k = 1, 2$) reflect the sampling weights on smaller entries in $A_{B_{ij}}$ in the first and second hidden layers, respectively. Our sampling strategy can be described as follows: at each iteration, for each submatrix $A_{B_{ij}}$,

(1) if $i > j$, each of the top[2] $d_1 \sqrt{\frac{d_i}{d_j}}$ largest edge weights $A_{ij}$ in $A_{B_{ij}}$ is sampled independently with probability $1 - p_{ij}^k$. The remaining entries in $A_{B_{ij}}$ are sampled independently with a probability of $p_{ij}^k$.

(2) if $i \leq j$, each of the top $d_1$ largest $A_{ij}$ in $A_{B_{ij}}$ are sampled with probability $1 - p_{ij}^k$. The remaining entries in $A_{B_{ij}}$ are sampled independently with a probability of $p_{ij}^k$.

---

[2]The values $d_1 \sqrt{\frac{d_i}{d_j}}$ and $d_1$ for selecting the top largest entries in $A_{ij}$ are chosen to simplify our theoretical analysis. In fact, any values in these orders are sufficient for our theoretical analysis. Note that the main idea of sampling edges between nodes of lower degrees with higher probability is preserved in our sampling strategy.

---

**Algorithm 1** SGD with Hierarchical Layer Sampling (HLS)

---

1: **Input:** Graph $\mathcal{G}$ with normalized adjancey matrix $A$, node features $X$, known labels in $\Omega$, step size $\eta_w$ and $\eta_v$, number of iterations $T$.
2: Initialize $W^{(0)}$, $V^{(0)}$, $C$. $\mathbf{W}_0 = 0$, $\mathbf{V}_0 = 0$
3: **for** $t = 0$ **to** $T - 1$ **do**
4:     Sample $A^{1t}$ using $p_{ij}^1$ and sample $A^{2t}$ using $p_{ij}^2$.
5:     Randomly sample $n$ from $\Omega$.
6:     Calculate the gradient of $L$ in (23) and update weight deviations through

$$\mathbf{W}_{t+1} \leftarrow \mathbf{W}_t - \eta_w \frac{\partial L(\mathbf{W},\mathbf{V})}{\partial \mathbf{W}}\Big|_{\mathbf{W}=\mathbf{W}_t,\mathbf{V}=\mathbf{V}_t},$$

$$\mathbf{V}_{t+1} \leftarrow \mathbf{V}_t - \eta_w \frac{\partial L(\mathbf{W},\mathbf{V})}{\partial \mathbf{V}}\Big|_{\mathbf{W}=\mathbf{W}_t,\mathbf{V}=\mathbf{V}_t}$$

7: **end for**
8: **Output:** $W^{(T)} = W^{(0)} + \mathbf{W}_T$, $V^{(T)} = V^{(0)} + \mathbf{V}_T$.

---

We allow the sampling weights to vary in different layers and will quantify how these weights affect generalization differently. When others are fixed, a small $p_{ij}^k$ indicates sampling primarily low-degree degree nodes, while a large $p_{ij}^k$ indicates sampling more edges between high-degree nodes.

To see why this sampling strategy prioritizes low-degree edges, consider the sampled edges between a fixed group $i$ and other groups $j$. Assume $p_{ij}^k$ are the same for all groups for simplicity. If $d_j < d_{j'}$ for two groups $j$ and $j'$, then $d_1\sqrt{\frac{d_i}{d_j}} > d_1\sqrt{\frac{d_i}{d_{j'}}}$, indicating that more lower-degree edges connecting groups $i$ and $j$ are sampled, compared with edges connecting groups $i$ and $j'$.

To analyze the impact of this graph topology sampling on the learning performance, we define the **sparse effective adjacency matrix** $A^*$ where in each submatrix $A^*_{B_{ij}}$:

(1) if $i > j$, the top $d_1\sqrt{\frac{d_i}{d_j}}$ largest values in $A_{B_{ij}}$ remain the same, while other entries are set to zero.

(2) if $i \le j$, the top $d_1$ largest values in $A_{B_{ij}}$ remain the same, while other entries are set to zero.

One can easily check from the definition that $\|A^*\|_1 = O(1)$, i.e., the maximum absolute column sum of $A^*$ is bounded by a constant. Moreover, $A^*$ is sparse by definition.

## 3.3 CONCEPT CLASS AND HIERARCHICAL LEARNING

Our theoretical generalization analysis shows that the returned GCN model achieves a prediction error that is a small constant times the minimum prediction error achieved by the best target function among a large concept class of functions. When the concept class characterizes the mapping function from node features to labels, the minimum prediction error by that class is close to zero, and so is the prediction error by the learned GCN model. Moreover, the concept class depends on the sparse $A^*$, rather than the dense $A$. As long as the sampled matrices are close $A^*$, despite being very sparse, graph sampling does not degrade generalization.

Consider a concept class of target functions $\mathcal{H}$ consisting of two smooth functions $\mathcal{F}$ and $\mathcal{G}$ : $\mathbb{R}^{d \times N} \times \mathbb{R}^{N \times N} \to \mathbb{R}^{k \times N}$ and a constant $\alpha \in \mathbb{R}^+$:

$$\mathcal{H}_{A^*}(X) = \mathcal{F}_{A^*}(X) + \alpha\mathcal{G}_{A^*}(\mathcal{F}_{A^*}(X)) \tag{6}$$

where the $r$th ($r \in [k]$) row of $\mathcal{F}_{A^*}$ and $\mathcal{G}_{A^*}$, denoted by $\mathcal{F}^r$ and $\mathcal{G}^r$: $\mathbb{R}^{d \times N} \times \mathbb{R}^{N \times N} \to \mathbb{R}^{1 \times N}$, satisfy

$$\mathcal{F}_{A^*}^r(X) = \sum_{i=1}^{p_{\mathcal{F}}} a_{\mathcal{F},r,i}^* \cdot \mathcal{F}^{r,i}\left(w_{r,i}^{*T} X A^*\right),$$

$$\mathcal{G}_{A^*}^r(X) = \sum_{i=1}^{p_{\mathcal{G}}} a_{\mathcal{G},r,i}^* \cdot \mathcal{G}^{r,i}\left(v_{r,i}^{*T} X A^*\right) \tag{7}$$

where $p_{\mathcal{F}}, p_{\mathcal{G}}$ are positive integers. Given $r, i$, the scalers $a^*_{\mathcal{F},r,i}, a^*_{\mathcal{G},r,i}$ are in $[-1, 1]$, the vectors $w^*_{r,i}$ are in $\mathbb{R}^d$, and $v^*_{r,i}$ are in $\mathbb{R}^k$. For simplicity, we assume $\left\|w^*_{r,i}\right\| = \left\|v^*_{r,i}\right\| = \frac{1}{\sqrt{2}}$ for all $r, i$. All $\mathcal{F}^{r,i}, \mathcal{G}^{r,i} : \mathbb{R} \to \mathbb{R}$ are smooth activation fuctions. We abuse notations in (7) to apply activations to every entry of a $1 \times N$ vector. The $n$th column of $\mathcal{H}_{A^*}$, denoted by $\mathcal{H}_{n,A^*} : \mathbb{R}^{d \times N} \times \mathbb{R}^{N \times N} \to \mathbb{R}^k$, represents the target function for node $n$.

$\mathcal{F}$ and $\mathcal{G}$ can both be viewed as one-hidde-layer GCNs with smooth activation functions and adjacent matrix $A^*$. The target function $\mathcal{H}$ includes the base signal $\mathcal{F}$, which is less complex yet contributes significantly to the target, and $\mathcal{G}$, which is more complicated but contributes less. Intuitively, we will show that using a two-hidden-layer GCN with a skip connection, the first hidden layer learns $\mathcal{F}$ and the second hidden layer learns $\mathcal{G}(\mathcal{F})$.

To quantify the complexity of target functions, we define model complexity constants $C_\varepsilon$ and sample complexity constants $C_{\mathfrak{s}}$, follow those in Li et al. (2022a) (Section 1.2) and Allen-Zhu & Li (2019) (Section 4).

Formally, for any smooth function $\phi(z)$ with its power series representation as $\phi(z) = \sum_{i=0}^{\infty} c_i z^i$, define two useful parameters as follows,

$$\mathcal{C}_\epsilon(\phi, R) = \sum_{i=0}^{\infty} \left( (C^*R)^i + \left( \frac{\sqrt{\log(1/\epsilon)}}{\sqrt{i}} C^*R \right)^i \right) |c_i| \tag{8}$$

$$\mathcal{C}_s(\phi, R) = C^* \sum_{i=0}^{\infty} (i+1)^{1.75} R^i |c_i| \tag{9}$$

where $R \geq 0$ and $C^*$ is a sufficiently large constant. $\mathfrak{C}_\varepsilon$ and $\mathfrak{C}_{\mathfrak{s}}$ represent the required number of model parameters and training samples to learn $\phi$ up to $\epsilon$ error, respectively. see Appendix B for more discussion.

We will show that the learned GCN by our method performs almost the same as the best function in the concept class in predicting unknown labels. Let $\mathcal{D}_{\tilde{x}_n}$ and $\mathcal{D}_{y_n}$ denote the distribution from which the feature and label of node $n$ are drawn, respectively. Let $\mathcal{D}$ denote the concatenation of these distributions. Then the given feature matrix $X$ and partial labels in $\Omega$ can be viewed as $|\Omega|$ identically distributed but correlated samples $(X, y_n)$ from $\mathcal{D}$. The correlation results from the fact that the label of node $i$ depends on not only the feature of node $i$ but also neighboring features.

To measure the label approximation performance of the target function, define

$$\min_{a^*_{\mathcal{F},r,i}, a^*_{\mathcal{G},r,i}, w^*_{r,i} v^*_{r,i}} \mathbb{E}_{n \in \mathcal{V}, (X,y_n) \sim \mathcal{D}} \left[ \frac{1}{2} \|\mathcal{H}_{n,A^*}(X) - y_n\|_2^2 \right] = \text{OPT} \tag{10}$$

as the minimum prediction error achieved by the best target function (over the choices of parameters $a^*_{\mathcal{F},r,i}, a^*_{\mathcal{G},r,i}, w^*_{r,i} v^*_{r,i}$) in the concept class. OPT decreases when the target functions are more complex, or the concept class enlarges, or if $A^*$ characterizes the node correlations properly.

### 3.4 MODELING THE PREDICTION ERROR OF UNKNOWN LABELS

To simplify the analysis and representation, we re-parameterize $U$ in (1) and (2) as $VC$, where $V \in \mathbb{R}^{m \times k}$. Then, (2) can be rewritten as follows:

$$\text{out}_n(X, A; W, V) = \text{out}_n^1(X, A) + C\sigma(V \text{out}^1(X, A)a_n)$$
$$\text{where } \text{out}^1(X, A; W) = C\sigma(WXA), \quad \text{out}_n^1(X, A; W) = C\sigma(WXa_n) \tag{11}$$

We follow the conventional setup for theoretical analysis that $C$ is fixed at its random initialization, and only $W$ and $V$ are updated during training. $C, W^{(0)}, V^{(0)}$ are randomly initialized from Gaussian distributions, $C_{i,j} \overset{\text{i.i.d.}}{\sim} \mathcal{N}\left(0, \frac{1}{m}\right)$, $W^{(0)}_{i,j} \overset{\text{i.i.d.}}{\sim} \mathcal{N}\left(0, \sigma_w^2\right)$ and $V^{(0)}_{i,j} \overset{\text{i.i.d.}}{\sim} \mathcal{N}\left(0, \sigma_v^2/m\right)$, respectively. $\sigma_w$ and $\sigma_v$ follow the parameter selection in Table 1.

The algorithm is summarized in Algorithm 1. When computing the stochastic gradient of a sampled label $y_n$, the loss is represented as a function of the weight deviations $\mathbf{W}, \mathbf{V}$ from initiation $W^{(0)}$ and $V^{(0)}$, i.e.,

$$L(\mathbf{W}, \mathbf{V}) = \text{Obj}_n(X, A^{1t}, A^{2t}, y_n; W^{(0)} + \mathbf{W}, V^{(0)} + \mathbf{V}). \tag{12}$$

$\mathbf{W}_t$ and $\mathbf{V}_t$ denote the weight deviations of the estimated weights $W^{(t)}$ and $V^{(t)}$ in iteration $t$ from $W^{(0)}$ and $V^{(0)}$, i.e., $W^{(t)} = W^{(0)} + \mathbf{W}_t, V^{(t)} = V^{(0)} + \mathbf{V}_t$. We assume $0 < \alpha \leq \widetilde{O}\left(\frac{1}{C_{\mathfrak{s}}(\mathcal{G}, \|A^*\|_1)}\right)$ throught the training. We prove that $\|\mathbf{W}_t\|_2$ and $\|\mathbf{V}_t\|_2$ are bounded by $\widetilde{\Theta}(C_{\mathfrak{s}}(\mathcal{F}))$ and $\widetilde{\Theta}(\alpha C_{\mathfrak{s}}(\mathcal{G}, \|A^*\|_1))$ during training, i.e., $\|\mathbf{W}_t\|_2 \leq \widetilde{\Theta}(C_{\mathfrak{s}}(\mathcal{F}))$, $\|\mathbf{V}_t\|_2 \leq \widetilde{\Theta}(\alpha C_{\mathfrak{s}}(\mathcal{G}, \|A^*\|_1)) < 1$ for all $t$. See Appendix C.5 for the proof.

The following lemma shows that graph sampling in different layers contributes to the output approximation differently. In other words, to maintain the same accuracy in the output, the tolerable sampling rates in different layers are different.

**Lemma 3.1** *For any given constant $E$, if the first and second layer $A^{1t}$ and $A^{2t}$ are sampled with $p_{ij}^1 \leq \widetilde{\Theta}(\frac{\sqrt{d_i d_j} E}{N_i N_j C_{\mathfrak{s}}(\mathcal{F}, \|A^*\|_1)})$ and $p_{ij}^2 \leq \widetilde{\Theta}(\frac{\sqrt{d_i d_j} E}{N_i N_j \alpha C_{\mathfrak{s}}(\mathcal{F}, \|A^*\|_1) C_{\mathfrak{s}}(\mathcal{G}, \|A^*\|_1)})$, respectively, then with the probability over $1 - e^{-\Omega(E\sqrt{d_i d_j}/C_{\mathfrak{s}}(\mathcal{F}))}$*

$$\left\|A^{1t} - A^*\right\|_1 \leq \frac{E}{\widetilde{\Theta}(C_{\mathfrak{s}}(\mathcal{F}))}, \text{ and } \left\|A^{2t} - A^*\right\|_1 \leq \frac{E}{\widetilde{\Theta}(\alpha C_{\mathfrak{s}}(\mathcal{F}, \|A^*\|_1) C_{\mathfrak{s}}(\mathcal{G}, \|A^*\|_1))}. \quad (13)$$

$$\left\|\text{out}_n\left(X, A^{1t}, A^{2t}; W^{(t)}, V^{(t)}\right) - \text{out}_n\left(X, A^*; W^{(t)}, V^{(t)}\right)\right\|_2 \leq E, \quad \text{for all } t. \quad (14)$$

Note that $\widetilde{\Theta}(\alpha C_{\mathfrak{s}}(\mathcal{G}, \|A^*\|_1)) < 1$, then the upper bound for $p_{ij}^2$ is higher than that for $p_{ij}^1$ in the assumption. That means the sampling for the first hidden layer must focus more on low-degree edges, while such a requirement is relaxed in the second layer. Then, (13) indicates that a larger deviation of the sampled matrix from $A^*$ can be tolerated in the second hidden layer compared with the first layer. Lemma 3.1 reveals that the skip connection allows for a more flexible sampling strategy in deeper layers.

We will show the learned model can achieve an error close to $O(\text{OPT})$. Our main theorem can be sketched as follows,

**Theorem 3.2** *For $\epsilon_0 = \widetilde{\Theta}(\alpha^4 C_{\mathfrak{s}}(\mathcal{G}, \|A^*\|_1)^4) < 1$ and $\epsilon = 10 \cdot \text{OPT} + \epsilon_0$, suppose sampling probability $p_{ij}^1 \leq \widetilde{\Theta}(\frac{\sqrt{d_i d_j} \epsilon_0}{N_i N_j C_{\mathfrak{s}}(\mathcal{F}, \|A^*\|_1)})$ and $p_{ij}^2 \leq \widetilde{\Theta}(\frac{\sqrt{d_i d_j} \epsilon_0}{N_i N_j \alpha C_{\mathfrak{s}}(\mathcal{F}, \|A^*\|_1) C_{\mathfrak{s}}(\mathcal{G}, \|A^*\|_1)})$, there exist $M_0 = \text{poly}(C_{\mathfrak{s}}(\mathcal{F}, \|A^*\|_1), C_{\mathfrak{s}}(\mathcal{G}, \|A^*\|_1), \|A^*\|_1, \alpha^{-1})$, $T_0 = \widetilde{\Theta}\left(\frac{C_{\mathfrak{s}}(\mathcal{F}, \|A^*\|_1)^2}{\|A^*\|_1 \min\{0.1, \epsilon^2\}}\right)$ and $N_0 = \widetilde{\Theta}(\Delta^4 C_{\mathfrak{s}}(\mathcal{F}, \|A^*\|_1)^2 \|A^*\|_1^4 \log N \epsilon^{-2})$ such that for every $m \geq M_0$, $T \geq T_0$ and $|\Omega| \geq N_0$, with high probability, the SGD algorithm satisfies*

$$\frac{1}{T} \sum_{t=0}^{T-1} \mathbb{E}_{n \in \mathcal{V}, (X, y_n) \sim \mathcal{D}} \left\|y_n - \text{out}_n\left(X, A^{1t}, A^{2t}; \mathbf{W}_t, \mathbf{V}_t\right)\right\|_2^2 \leq \epsilon \quad (15)$$

As a sanity check, when the concept class enlarges, OPT decreases. Theorem 3.2 shows that the required number of neurons $M_0$ (model complexity) and labels $N_0$ (sample complexity) both increase accordingly. $p_{ij}^1$ and $p_{ij}^2$ decreasing means that we should sample more high-weight edges. Thus, our theoretical bounds match the intuition that a larger model, more labels, and more high-weight edges improve the prediction accuracy. $N_0$ is in the order of $\log N$, indicating that the unknown labels can be accurately predicted from partial labels. Moreover, when $\|A^*\|_1$ increases, $C_{\varepsilon}$ and $C_{\mathfrak{s}}$, and $\epsilon_0$ all increase. Theorem 3.2 indicates the model complexity $M_0$, $N_0$, and the generalization error $\epsilon$ all increasing, indicating worse prediction performance.

This proof of Theorem 3.2 builds upon the proof of Theorem 1 in Allen-Zhu & Li (2019), which analyzes the generalization of a three-layer ResNet for a supervised regression problem. We extend the analysis to training GCNs with graph sampling for a semi-supervised node regression problem. The main technical challenge is to handle the dependence of labels on neighboring features and the error in adjacency matrices due to the sampling. Compared with Li et al. (2022a) which also considers training GCN with graph sampling, we consider a different sampling method from that in Li et al. (2022a). The resulting $A^*$ in Li et al. (2022a) is a dense matrix as $A$, while $A^*$ in our paper is a sparse matrix. Our results thus allow the sampled matrices to be very sparse while still maintaining the generalization accuracy. Moreover, the sampling method is the same for both hidden layers in Li et al. (2022a), resulting the same deviation from $A^*$ in both layers. Our results indicate that the skip connection allows a more flexible sampling method in the second layer.

## 4 EMPIRICAL EXPERIMENT

### 4.1 EXPERIMENT ON SYNTHETIC DATA

We generate a graph $\mathcal{G}$ with $N = 2000$ nodes. $\mathcal{G}$ has two-degree groups. Group 1 has $N_1 = 200$ nodes, and the degree of each node follows a Gaussian distribution $\mathcal{N}\left(d_1, \sigma^2\right)$. Group 2 has $N_2 = 1800$ nodes, and the degree of each node follows a Gaussian distribution $\mathcal{N}\left(d_2, \sigma^2\right)$. $d_1 = 200$ and $\sigma = 20$. The degrees are truncated to fall within the range of 0 to 500. Given $A$, the sparse $A^*$ is obtained following the procedure in Section 3.2. The node labels are generated by the target function

$$\mathcal{F}_{A^*}(X) = CW^* X A^* \tag{16}$$

$$\mathcal{G}_{A^*}(\mathcal{F}_{A^*}(X)) = C\left(\sin\left(V^*\mathcal{F}_{A^*}(X)A^*\right) \odot \tanh\left(V^*\mathcal{F}_{A^*}(X)A^*\right)\right) \tag{17}$$

$$\mathcal{H}_{A^*}(X) = \mathcal{F}_{A^*}(X) + \alpha \mathcal{G}_{A^*}(\mathcal{F}_{A^*}(X)) \tag{18}$$

where $X \in \mathbb{R}^{d \times N}$, $W^* \in \mathbb{R}^{r \times d}$, $V^* \in \mathbb{R}^{r \times k}$, and $C \in \mathbb{R}^{k \times r}$ are randomly generated with each entry i.i.d. from $\mathcal{N}(0,1)$. $d = 100$, $k = 5$, $r = 30$, $\alpha = 0.5$. A two-hidden-layer GCN with a single skip-connection as defined in (2) with $m$ neurons in each hidden layer is trained on a randomly selected set $\Omega$ of labeled nodes. The rest $N - |\Omega|$ labels are used for testing. The test error is measured by the $\ell_2$ regression loss in (3).

**Model and sample complexities with** $\|A^*\|_1$: In Figures 1 (a) (b), we vary $A^*$ by changing $d_2$ and the corresponding $A$. We directly train with $A^*$ to study the impact of $A^*$ on model and sample complexities. In Figures 1 (a), $|\Omega| = 1200$ and the number of neurons per layer $m$ varies. To achieve the same test accuracy, when $\|A^*\|_1$ increases, the number of neurons also increases, verifying our model complexity $M_0$ in Theorem 3.2. In Figures 1 (b), $m = 50$ and $|\Omega|$ varies. To achieve the same test accuracy, when $\|A^*\|_1$ increases, the required number of labels also increases, verifying our sample complexity $N_0$ in Theorem 3.2.

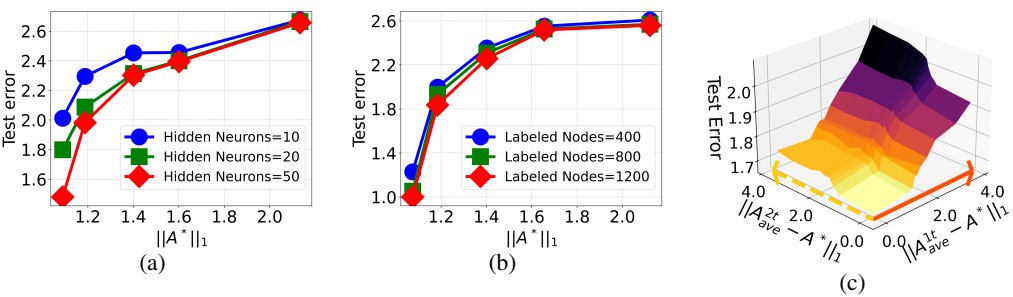

(a)  (b)  (c)

Figure 1: Experiment on synthetic data (a) Test error with model complexity and $\|A^*\|_1$. (b) Test error with sample complexity and $\|A^*\|_1$. (c) The second hidden layer tolerates more error than the first hidden layer. See more details in Appendix D.

**Layer-wise sampling impact on generalization**: We fix $\Omega = 1200$, $m = 50$, $\|A^*\|_1 = 1.27$. We sample adjacency matrices during training. Figure 1 (c) shows the relationship between the test error and the average deviation of sampled matrices ($A^{1t}$ and $A^{2t}$ in the first and second hidden layers) from $A^*$. One can see that sampling in the second hidden layer (yellow dashed line) contributes to generalization degradation much milder than sampling in the first hidden layer (red solid arrow). This verifies our Lemma 3.1 that the sampling requirements are more restrictive in the first layer than the second layer to maintain the same generalization accuracy.

### 4.2 EXPERIMENT ON REAL DATA

We evaluate on the large-scale Open Graph Benchmark (OGB) datasets for node multi-class classification tasks. The results on Ogbn-Arxiv are reported here. The results on Ogbn-Products and details of the datasets are in Appendix D. The task of Ogbn-Arxiv is to classify the 40 subject areas of arXiv CS papers. We use 60% of the data for training, 20% for testing, and 20% for verification. We deploy an 8-layer Jumping Knowledge Network (Xu et al., 2018b) GCN with concatenation layer aggregation as a learner network. Each hidden layer consists of 128 neurons. We treat the first four layers as shallow layers and the last four layers as deep layers. Shallow and deep layers are sampled

differently. The generalization is evaluated by the fraction of erroneous predictions of unknown labels.

**Sampling in deep layers is more flexible with less generalization degradation**. In this experiment, we employ a simplified version of the graph sampling method discussed in Section 3.2. For the shallow layers, at each iteration $t$, we obtain a sampled adjacency matrix $A^{1t}$ as follows: we sample the top $q_1$ fraction of the largest edge weights $A_{ij}$ from the adjacency matrix $A$ with a 99% probability, and sample the remaining entries with a 1% probability. For the deep layers, the sampled adjacency matrix $A^{2t}$ is generated similarly, but we use the top $q_2$ fraction of largest $A_{ij}$, again sampling with probabilities of 99% and 1% for the top and remaining entries, respectively.

Figure 2 (a) shows test error when $q_1$ and $q_2$ vary. One can see that the test error decreases more drastically when only increasing $q_1$ (yellow dashed arrow) compared with only increasing $q_2$ (red solid arrow), indicating that graph sampling in shallow layers has a more significant impact than graph sampling in deeper layers. When both $q_1$ and $q_2$ are greater than 0.6, the test error is always small (less than 0.29) for a wide range of $q_1, q_2$. That may suggest the existence of multiple sparse $A^*$ such that sampled matrices with different $q_1, q_2$ pairs approximate different $A^*$, and all $A^*$ can accurately represent the data correlations in the mapping function from the features to the labels.

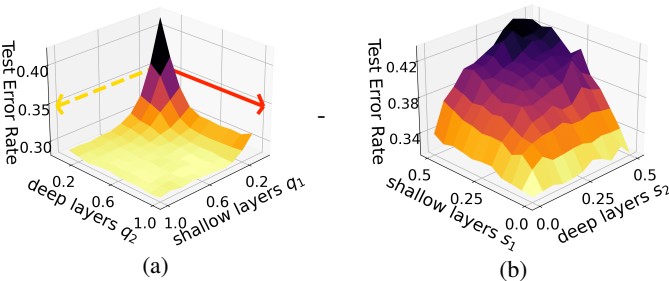

$$(a) \qquad\qquad\qquad (b)$$

Figure 2: Learning deep GCNs on Ogbn-Arxiv (a) Deeper layers tolerate higher sampling rates than shallow layers while maintaining accuracy. (b) Sampling more low-degree edges (small $s_1, s_2$) outperforms sampling more high-degree edges (large $s_1, s_2$). See 2D projection in Figure 4.

**Lower-degree edges are more influential on generalization than high-degree edges**. Note that if nodes $i$ and $j$ have higher degrees, then $A_{ij}$ has a smaller value. We sample one matrix for the shallow layers by keeping the values of $A_{ij}$ that are in the range of top $s_1$ to $s_1 + 0.5$ fraction and setting all other values to zero. Similarly, we sample one matrix for the deep layers by keeping the values in the range of top $s_2$ to $s_2 + 0.5$ fraction and setting all other values to zero. These two sampled matrices are used during training. When $s_1$ and $s_2$ increase, the resulting matrices have the same number of nonzero entries, and the sampled entries focus more on high-degree edges. Figure 2 (b) shows the test error indeed increases as $s_1, s_2$ increase. This justifies the sampling strategy to sample more low-degree edges.

## 5 CONCLUSION

This paper provides a theoretical generalization analysis of training GCNs with skip connections using graph sampling. To the best of our knowledge, this paper provides the first analysis of how skip connection affects the generalization performance. We show that for a two-hidden-layer GCN with a skip connection, the first hidden layer learns a simpler function that contributes significantly to the output, making the choice of sampling more crucial in the first hidden layer. In contrast, the second layer learns a composite function that contributes less to the output, allowing for a more flexible sampling approach while preserving generalization. This insight is verified on deep GCNs on benchmark datasets. Future works include generalization analysis of deep GCNs and designing layer-specific sampling strategies that optimize generalization within the constraints of available computational resources.

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

## A  PRELIMINARIES

**Lemma A.1** *If $M \in \mathbb{R}^{n \times m}$ is a random matrix where $M_{i,j}$ are i.i.d. from $\mathcal{N}(0,1)$. Then,*

- *For any $t \geq 1$, with probability $1 - e^{-t^2}$, it satisfies*

$$\|M\|_2 \leq O\left(\sqrt{n} + \sqrt{m}\right) + t.$$

- *If $1 \leq s \leq O\left(\frac{m}{\log^2 m}\right)$, then with probability $1 - e^{-(n+s\log^2 m)}$ it satisfies*

$$\|Mv\|_2 \leq O\left(\sqrt{n} + \sqrt{s \log m}\right) \|v\|_2$$

*for all $s$-sparse vectors $v \in \mathbb{R}^m$.*

Proof: The statement can be found in Proposition B.2. from Allen-Zhu & Li (2019)

**Lemma A.2** *Suppose $\delta \in [0,1]$ and $g^{(0)} \in \mathbb{R}^m$ is a random vector $g^{(0)} \sim \mathcal{N}(0, I_m)$. With probability at least $1 - e^{-\Omega(m\delta^{2/3})}$, for all vectors $g' \in \mathbb{R}^m$ with $\|g'\|_2 \leq \delta$, letting $D' \in \mathbb{R}^{m \times m}$ be the diagonal matrix where $(D')_{k,k} = \mathbf{1}_{(g^{(0)}+g')_k} - \mathbf{1}_{(g^{(0)})_k}$ for each $k \in [m]$, we have*

$$\|D'\|_0 \leq O(m^{2/3}) \quad \text{and} \quad \|D'g^{(0)}\|_2 \leq \|g'\|_2.$$

Proof: The statement can be found in Proposition B.4. from Allen-Zhu & Li (2019)

**Lemma A.3** *Given a sampling set $X = \{x_n\}_{n=1}^N$ that contains $N$ partly dependent random variables, for each $n \in [N]$, suppose $x_n$ is dependent with at most $d_X$ random variables in $X$ (including $x_n$ itself), and the moment generating function of $x_n$ satisfies $\mathbb{E}[e^{sx_n}] \leq e^{Cs^2}$ for some constant $C$ that may depend on $x_n$. Then, the moment generation function of $\sum_{n=1}^N x_n$ is bounded as*

$$\mathbb{E}[e^{s\sum_{n=1}^N x_n}] \leq e^{Cd_X N s^2}.$$

Proof: The statement can be found in Lemma 7 from Zhang et al. (2020b)

**Lemma A.4** $\|Xa_n\| \leq \|A\|_1$.

Proof:

$$
\begin{aligned}
\|Xa_n\| &= \|\sum_{k=1}^N x_k a_{k,n}\| \\
&= \|\sum_{k=1}^N \frac{a_{k,n}}{\sum_{k=1}^N a_{k,n}} x_k\| \cdot \sum_{k=1}^N a_{k,n} \\
&\leq \sum_{k=1}^N \frac{a_{k,n}}{\sum_{k=1}^N a_{k,n}} \|x_k\| \cdot \|A\|_1 \\
&= \|A\|_1
\end{aligned}
\tag{19}
$$

**Lemma A.5** *If $\mathcal{F} : \mathbb{R}^d \to \mathbb{R}^k$ has general complexity $(p, \mathfrak{C}_{\mathfrak{s}}(\mathcal{F}), \mathfrak{C}_{\varepsilon}(\mathcal{F}))$, then for every $x, y \in \mathbb{R}^d$, it satisfies $\|\mathcal{F}(x)\|_2 \leq \sqrt{k}p\mathfrak{C}_{\mathfrak{s}}(\mathcal{F}) \cdot \|x\|_2$ and $\|\mathcal{F}(x) - \mathcal{F}(y)\|_2 \leq \sqrt{k}p\mathfrak{C}_{\mathfrak{s}}(\mathcal{F}) \cdot \|x - y\|_2$.*

Proof: The boundedness of $\|\mathcal{F}(x)\|_2$ is trivial so we only focus on $\|\mathcal{F}(x) - \mathcal{F}(y)\|_2$. For each component $g(x) = \mathcal{F}_{r,i}\left(\frac{\langle w_{1,i}^*, (x,1)\rangle}{\|(x,1)\|_2}\right) \cdot \langle w_{2,i}^*, (x,1)\rangle$, denoting by $w_1^*$ as the first d coordinate of $w_{1,i}^*$, and by $w_{2,i}^*$ as the first d coordinates of $w_{2,i}^*$, we have

$$g'(x) = \mathcal{F}_{r,i}\left(\frac{\langle w_{1,i}^*, (x,1)\rangle}{\|(x,1)\|_2}\right) \cdot w_2^*$$

$$+ \langle w_{2,i}^*, (x,1)\rangle \cdot \mathcal{F}_{r,i}'\left(\frac{\langle w_{1,i}^*, (x,1)\rangle}{\|(x,1)\|_2}\right) \cdot \frac{w_1^* \cdot \|(x,1)\|_2 - \langle w_{1,i}^*, (x,1)\rangle \cdot (x,1)/\|(x,1)\|_2^2}{\|(x,1)\|_2^2}$$

This implies

$$\|g'(x)\|_2 \le \left|\mathcal{F}_{r,i}\left(\frac{\langle w_{1,i}^*, (x,1)\rangle}{\|(x,1)\|_2}\right)\right| + 2\left|\mathcal{F}_{r,i}'\left(\frac{\langle w_{1,i}^*, (x,1)\rangle}{\|(x,1)\|_2}\right)\right| \le 3\mathfrak{C}_{\mathfrak{s}}\left(\mathcal{F}_{r,i}\right)$$

As a result, $|\mathcal{F}_r(x) - \mathcal{F}_r(y)| \le 3p\mathfrak{C}_{\mathfrak{s}}\left(\mathcal{F}_{r,i}\right)$.

**Lemma A.6** *For every smooth function $\phi$, every $\epsilon \in (0, \frac{1}{\mathcal{C}(\phi,a)\sqrt{a^2+1}})$, there exists a function $h : \mathbb{R}^2 \to [-\mathcal{C}_\epsilon(\phi,a)\sqrt{a^2+1}, \mathcal{C}_\epsilon(\phi,a)\sqrt{a^2+1}]$ that is also $\mathcal{C}_\epsilon(\phi,a)\sqrt{a^2+1}$-Lipschitz continuous on its first coordinate with the following two (equivalent) properties:*
*(a) For every $x_1 \in [-a, a]$ where $a > 0$:*

$$\left|\mathbb{E}\left[\mathbf{1}_{\alpha_1 x_1 + \beta_1\sqrt{a^2-x_1^2}+b_0 \ge 0} h(\alpha_1, b_0)\right] - \phi(x_1)\right| \le \epsilon$$

*where $\alpha_1, \beta_1, b_0 \sim \mathcal{N}(0,1)$ are independent random variables.*
*(b) For every $\mathbf{w}^*, \mathbf{x} \in \mathbb{R}^d$ with $\|\mathbf{w}^*\|_2 = 1$ and $\|\mathbf{x}\| \le a$:*

$$\left|\mathbb{E}\left[\mathbf{1}_{\mathbf{wX}+b_0 \ge 0} h(\mathbf{w}^\top \mathbf{w}^*, b_0)\right] - \phi(\mathbf{w}^{*\top}\mathbf{x})\right| \le \epsilon$$

*where $\mathbf{w} \sim \mathcal{N}(0,\mathbf{I})$ is an $d$-dimensional Gaussian, $b_0 \sim \mathcal{N}(0,1)$.*
*Furthermore, we have $\mathbb{E}_{\alpha_1, b_0 \sim \mathcal{N}(0,1)}[h(\alpha_1, b_0)^2] \le (\mathcal{C}_s(\phi,a))^2(a^2+1)$.*
*(c) For every $\mathbf{w}^*, \mathbf{x} \in \mathbb{R}^d$ with $\|\mathbf{w}^*\|_2 = 1$, let $\tilde{\mathbf{w}} = (\mathbf{w}, b_0) \in \mathbb{R}^{d+1}$, $\tilde{\mathbf{x}} = (\mathbf{x}, 1) \in \mathbb{R}^{d+1}$ with $\|\tilde{\mathbf{x}}\| \le \sqrt{a^2+1}$, then we have*

$$\left|\mathbb{E}\left[\mathbf{1}_{\tilde{\mathbf{w}}^\top \tilde{\mathbf{x}} \ge 0} h(\tilde{\mathbf{w}}[1:d]^\top \mathbf{w}^*, \tilde{\mathbf{w}}[d+1])\right] - \phi(\mathbf{w}^{*\top}\tilde{\mathbf{x}}[1:d])\right| \le \epsilon$$

*where $\tilde{\mathbf{w}} \sim \mathcal{N}(0, \mathbf{I}_{d+1})$ is an $d$-dimensional Gaussian.*
*We also have $\mathbb{E}_{\tilde{\mathbf{w}} \in \mathcal{N}(0,\mathbf{I}_{d+1})}[h(\tilde{\mathbf{w}}[1:d]^\top \mathbf{w}^*, \tilde{\mathbf{w}}[d+1])^2] \le (\mathcal{C}_s(\phi,a))^2(a^2+1)$.*

Proof: The statement can be found in Lemma B.1. from Li et al. (2022a)

## B CONCEPT CLASS

When the target function is more complex, $\mathcal{H}_n$ will approach $y_n$, and $OPT$ will be closer to zero. To evaluate the complexity of the target function, we introduce model complexity constants $\mathfrak{C}_\varepsilon$ and sample complexity constants $\mathfrak{C}_\mathfrak{s}$ in (8) and (9) following the definitions in (Allen-Zhu & Li, 2019; Li et al., 2022a). Many population functions have bounded complexity. For instance, if $\phi(z)$ is $\exp(z)$, $\sin(z)$, $\cos(z)$ or polynomials of $z$, then $\mathcal{C}_\epsilon(\phi, O(1)) \le O(\text{poly}(1/\epsilon))$ and $\mathcal{C}_s(\phi, O(1)) \le O(1)$.

We define $\mathfrak{C}_\varepsilon(\mathcal{F}, \|A^*\|_1) = \max_{r,i}\{\mathfrak{C}_\varepsilon\left(\mathcal{F}^{r,i}, \|A^*\|_1\right)\}$ and $\mathfrak{C}_\mathfrak{s}(\mathcal{F}, \|A^*\|_1) = \max_{r,i}\{\mathfrak{C}_\mathfrak{s}\left(\mathcal{F}^{r,i}, \|A^*\|_1\right)\}$ as the model and sample complexity of $\mathcal{F}$, respectively, and similarly define $\mathfrak{C}_\varepsilon(\mathcal{G}, \|A^*\|_1)$ and $\mathfrak{C}_\mathfrak{s}(\mathcal{G}, \|A^*\|_1)$ for $\mathcal{G}$.

$\mathcal{F}$ and $\mathcal{G}$ are composed by $p_\mathcal{F}$ and $p_\mathcal{G}$ different smooth functions. The complexity of $\mathcal{F}$ is the most complex sub-target function $\mathcal{F}^{r,i}$ in the $p_\mathcal{F}$ functions.

We define $\mathfrak{B}_\mathcal{F} := \max_n \|\mathcal{F}_{n,A^*}(X)\|_2$, $\mathfrak{B}_{\mathcal{F}\circ\mathcal{G}} := \max_n \|\mathcal{G}_{n,A^*}(\mathcal{F}_{A^*}(X))\|_2$ for all $X$ satisfying $\|x_n\| = 1$. Assume $\mathcal{G}(\cdot)$ is $\mathfrak{L}_\mathcal{G}$ Lipschitz continuous. It is simple to verify (see Lemma A.5) that $\mathfrak{B}_\mathcal{F} \le \sqrt{k}p_\mathcal{F}\mathfrak{C}_\mathfrak{s}(\mathcal{F}, \|A^*\|_1)\|A^*\|_1$, $\mathfrak{L}_\mathcal{G} \le \sqrt{k}p_\mathcal{G}\mathfrak{C}_\mathfrak{s}(\mathcal{G}, \mathfrak{B}_\mathcal{F}\|A^*\|_1)$ and $\mathfrak{B}_{\mathcal{F}\circ\mathcal{G}} \le kp_\mathcal{F}\mathfrak{C}_\mathfrak{s}(\mathcal{F}, \|A^*\|_1) \cdot p_\mathcal{G}\mathfrak{C}_\mathfrak{s}(\mathcal{G}, \mathfrak{B}_\mathcal{F}\|A^*\|_1)\mathfrak{B}_\mathcal{F}\|A^*\|_1$.

## C  THEOREM 3.2 PROOF DETAILS

Let us define learner networks that are single-skip two-hidden-layer with ReLU activation $\text{out}_n$ : $\mathbb{R}^{d \times N} \times \mathbb{R}^N \to \mathbb{R}^k$ with

$$\text{out}_n(X, A; W, V) = \text{out}_n^1(X, A) + C\sigma(V \text{out}^1(X, A)a_n) \tag{20}$$

and

$$\text{out}^1(X, A) = CD_W \odot (WXA^1) \in R^{k \times N}, \tag{21}$$

$$\text{out}_n^1(X, A) = CD_W^n WXa_n^1 \in R^k \tag{22}$$

where

- $A = [a_1, \quad a_2, \quad \cdots, \quad a_N] \in R^{N \times N}$ denotes the normalized adjacency matrix.
- $X \in R^{d \times N}$ denotes the the matrix of $d$ dimension features of $N$ nodes.
- $W \in R^{m \times d}$ denotes the first hidden layer weight.
- $V \in R^{m \times k}$ denotes the second hidden layer weight.
- $C \in R^{k \times m}$ denotes the output layer weight.
- $D_W = \begin{bmatrix} \mathbf{1}_{(WXa_1 \geq 0)}, & \mathbf{1}_{(WXa_2 \geq 0)}, & \cdots & , \mathbf{1}_{(WXa_N \geq 0)} \end{bmatrix}, D_W^n = \text{diag}\{\mathbf{1}_{(WXa_n \geq 0)}\}$
- $D_V = \begin{bmatrix} \mathbf{1}_{(V \text{out}_n^1(X,A)a_1 \geq 0)} & \mathbf{1}_{(V \text{out}_n^1(X,A)a_2 \geq 0)} & \cdots & \mathbf{1}_{(V \text{out}_n^1(X,A)a_N \geq 0)} \end{bmatrix}, D_V^n = \text{diag}\{\mathbf{1}_{(V \text{out}_n^1(X,A)a_n \geq 0)}\}$

The $l_2$ loss is represented as a function of the weight deviations $\mathbf{W}, \mathbf{V}$ from initiation $W^{(0)}$ and $V^{(0)}$, i.e.,

$$L(\mathbf{W}, \mathbf{V}) = \text{Obj}_n(X, A^{1t}, A^{2t}, y_n; W^{(0)} + \mathbf{W}, V^{(0)} + \mathbf{V}). \tag{23}$$

Let $W^{(t)} = W^{(0)} + \mathbf{W}_t, V^{(t)} = V^{(0)} + \mathbf{V}_t$. We assume $0 < \alpha \leq \widetilde{O}\left(\frac{1}{kp_{\mathcal{G}}\mathfrak{C}_{\mathfrak{s}}(\mathcal{G}, \mathfrak{B}_{\mathcal{F}}\|A^*\|_1)}\right)$ throught the training. We prove that $\|\mathbf{W}_t\|_2$ and $\|\mathbf{V}_t\|_2$ are bounded by $\tau_w$ and $\tau_v$ in Table 1 during training, i.e., $\|\mathbf{W}_t\|_2 \leq \tau_w, \|\mathbf{V}_t\|_2 \leq \tau_v$ for all $t$. See Appendix C.5 for the proof.

Table 1: Parameter choices

| $\sigma_w$ | $m^{-\frac{1}{2}+0.01} \leq \sigma_w \leq m^{-0.01}$ | $\sigma_v$ | $\sigma_v = \Theta(\text{polylog}(m))$ |
|---|---|---|---|
| $\tau_w$ | $\tau_w = \widetilde{\Theta}(kp_{\mathcal{F}}\mathfrak{C}_{\mathfrak{s}}(\mathcal{F}, \|A^*\|_1))$ and $m^{\frac{1}{8}+0.001}\sigma_w \leq \tau_w \leq m^{\frac{1}{8}-0.001}\sigma_w^{\frac{1}{4}}$ | | |
| $\tau_v$ | $\tau_v = \widetilde{\Theta}(\alpha kp_{\mathcal{G}}\mathfrak{C}_{\mathfrak{s}}(\mathcal{G}, \mathfrak{B}_{\mathcal{F}}\|A^*\|_1)\mathfrak{B}_{\mathcal{F}}\|A^*\|_1)$ and $\sigma_v\left(\frac{k}{m}\right)^{\frac{3}{8}} \leq \tau_v \leq \frac{\sigma_v}{\text{polylog}(m)\|A^*\|_1} < 1$ | | |

### C.1  PROOF OVERVIEW

In practice, for computational efficiency, we use the sampled adjacency matrix $A^t$ in the learning network. Therefore, we should consider the discrepancy between the target function with $A^*$ and the practical learning network:

$$\begin{aligned} \|\mathcal{H}_{n,A^*}(X) - \text{out}_n(X, A^{1t}, A^{2t}; \mathbf{W}_t, \mathbf{V}_t)\|_2 &\leq \|\mathcal{H}_{n,A^*}(X) - \text{out}_n(X, A^*)\|_2 \\ &+ \|\text{out}_n(X, A^{1t}, A^{2t}) - \text{out}_n(X, A^*)\|_2. \end{aligned} \tag{24}$$

First, we prove the existence of $\mathbf{W}^*$ and $\mathbf{V}^*$ with $m \geq M_0$, $T \geq T_0$ and $\Omega \geq N_0$ such that

$$\|\mathcal{H}_{n,A^*}(X) - \text{out}_n(X, A^*; \mathbf{W}^*, \mathbf{V}^*)\|_2 \leq \epsilon_0. \tag{25}$$

Second, we demonstrate that

$$\begin{aligned} \|\text{out}_n(X, A^{1t}, A^{2t}) - \text{out}_n(X, A^*)\|_2 &\leq \|C\sigma(WXa_n^{1t}) - C\sigma(WXa_n^*)\|_2 \\ &+ \|C\sigma(V \text{out}_n^1(XA^{1t})a_n^{2t}) - C\sigma(V \text{out}_n^1(XA^*)a_n^*)\|_2. \end{aligned}$$

For the inequality $\left\|C\sigma(WXa_n^{1t}) - C\sigma(WXa_n^*)\right\|_2 \le \epsilon_0$ to hold, it is required that $\left\|A^{1t} - A^*\right\|_1 \le \frac{\epsilon_0}{\widetilde{\Theta}(C_s(\mathcal{F}))}$ and similarly, $\left\|A^{2t} - A^*\right\|_1 \le \frac{\epsilon_0}{\widetilde{\Theta}(\alpha C_s(\mathcal{F})C_s(\mathcal{G},\|A^*\|_1))}$.

Third, we establish that with appropriate sampling probabilities $p_{ij}^1$ and $p_{ij}^2$, the norms $\left\|A^{1t} - A^*\right\|_1$ and $\left\|A^{2t} - A^*\right\|_1$ can be sufficiently small.

Finally, consider the definition of OPT, we can prove $\left\|y_n - \text{out}_n\left(X, A^{1t}, A^{2t}; \mathbf{W}_t, \mathbf{V}_t\right)\right\|_2 \le \epsilon$.

## C.2 COUPLING

**Lemma C.1** *We show that the weights after a properly bounded amount of updates stay close to the initialization, and many good properties occur. Suppose that $\|\mathbf{W}\|_2 \le \tau_w$, $\|\mathbf{V}\|_2 \le \tau_v$, $W_0$ from $\mathcal{N}\left(0, \sigma_w^2\right)$ and $V_0$ from $\mathcal{N}\left(0, \sigma_v^2/m\right)$, we have that*

*1.*
$$\left\|D_{\mathbf{W}^{(0)}}^n - D_{\mathbf{W}+\mathbf{W_0}}^n\right\|_0 \le O\left((\frac{\tau_w}{\sigma_w})^{2/3}m^{2/3}\right) \tag{26}$$

*2.*
$$\left\|\mathbf{C}D_{\mathbf{W}+\mathbf{W_0}}^n\mathbf{W}Xa_n - \mathbf{C}D_{\mathbf{W}+\mathbf{W_0}}^n\left(\mathbf{W}+\mathbf{W}_0\right)Xa_n\right\|_2 \le \widetilde{O}\left(\frac{\sqrt{s}}{\sqrt{m}}\tau_w\|A\|_1 + \sqrt{k}\sigma_w\|A\|_1\right) \tag{27}$$

*3.*
$$\left\|\text{out}_n^1(X, A; W)\right\|_2 \le \widetilde{O}(\tau_w\|A\|_1) \tag{28}$$

*4.*
$$\left\|D_{\mathbf{V}^{(0)}}^n - D_{\mathbf{V}+\mathbf{V_0}}^n\right\|_0 \le O\left((\frac{\tau_v}{\sigma_v})^{2/3}m\right) \tag{29}$$

*5.*
$$\left\|\mathbf{C}D_{\mathbf{V}+\mathbf{V_0}}^n\mathbf{V}\,\text{out}^1(X, A)a_n - \mathbf{C}D_{\mathbf{V}+\mathbf{V_0}}^n\left(\mathbf{V}+\mathbf{V}_0\right)\text{out}^1(X, A)a_n\right\|_2$$
$$\le \widetilde{O}\left((\frac{\sqrt{k}}{\sqrt{m}}\sigma_v + \frac{\sqrt{s}}{\sqrt{m}}\tau_v)\|\text{out}_n^1(X, A)\|_2\|A\|_1\right) \tag{30}$$

*6.*
$$\left\|\mathbf{C}D_{\mathbf{V}+\mathbf{V_0}}^n\mathbf{V}_0\right\|_2 \le \tau_v(\frac{\tau_v}{\sigma_v})^{1/3} \tag{31}$$

*7.*
$$\left\|\mathbf{C}D_{\mathbf{V}+\mathbf{V_0}}^n\left(\mathbf{V}+\mathbf{V}_0\right)\text{out}^1(X, A)a_n\right\|_2 \le \widetilde{O}\left(\tau_v\|\text{out}_n^1(X, A)\|_2\|A\|_1\right) \tag{32}$$

Proof:

1. $\|WXa_n\|_2 \le \|W\|_2\|Xa_n\|_2 \le \tau_w\|Xa_n\|_2$ and $\langle W_0, Xa_n\rangle_j \sim \mathcal{N}(0, \|Xa_n\|_2^2\sigma_w^2)$, using Lemma A.2, we have
$$\left\|D_{\mathbf{W}^{(0)}}^n - D_{\mathbf{W}+\mathbf{W_0}}^n\right\|_0 \le O\left((\frac{\tau_w\|Xa_n\|_2}{\sigma_w\|Xa_n\|_2\sqrt{m}})^{2/3}m\right) \tag{33}$$

2. We write $\mathbf{C}D_{\mathbf{W}+\mathbf{W_0}}^n\mathbf{W}Xa_n^{*1} - \mathbf{C}D_{\mathbf{W}+\mathbf{W_0}}^n\left(\mathbf{W}+\mathbf{W}_0\right)Xa_n^{*1} = -\mathbf{C}D_{\mathbf{W_0}}^n\mathbf{W}^{(0)}Xa_n^{*1} + \mathbf{C}\left(D_{\mathbf{W_0}}^n - D_{\mathbf{W}+W_0}^n\right)\mathbf{W}_0Xa_n^{*1}$. For the first term, $\left\|D_{\mathbf{W_0}}^n\mathbf{W}_0Xa_n\right\|_2 \le \|\mathbf{W}_0Xa_n\|_2 \le O\left(\sigma_w\|A\|_1\sqrt{m}\right)$, so $\left\|CD_{\mathbf{W_0}}^n\mathbf{W}_0Xa_n\right\|_2 \le \widetilde{O}\left(\sqrt{k}\sigma_w\|A\|_1\right)$

   For the second term, using Lemma A.2 again, we have
   $$\left\|\left(D_{\mathbf{W_0}}^n - D_{\mathbf{W}+W_0}^n\right)\mathbf{W}_0Xa_n\right\|_2 \le \|WXa_n\|_2 \le \tau_w\|A\|_1$$

   Using Lemma A.1, for every $s$-sparse vector $\mathbf{y}$, it satisfies $\|\mathbf{A}\mathbf{y}\|_2 \le e^{O(\sqrt{\frac{s}{m}})}\|\mathbf{y}\|_2$ with high probability. The sparsity of the second term is $s = (\frac{\tau_w}{\sigma_w\|A\|_1})^{2/3}m^{2/3}$, so we have
   $$\left\|\mathbf{C}\left(D_{\mathbf{W_0}}^n - D_{\mathbf{W}+W_0}^n\right)\mathbf{W}_0Xa_n\right\|_2 \le \widetilde{O}\left(\frac{\sqrt{s}}{\sqrt{m}}\right)\cdot\|\mathbf{W}Xa_n\|_2 \le \widetilde{O}\left(\frac{\sqrt{s}}{\sqrt{m}}\tau_w\|A\|_1\right).$$

3. $\left\|\mathrm{out}_n^1(X, A)\right\|_2 \leq \left\|\mathbf{C}D_{\mathbf{W}+\mathbf{W_o}}^n \mathbf{W} X a_n\right\|_2 + \left\|\mathbf{C}D_{\mathbf{W}+\mathbf{W_o}}^n \mathbf{W} X a_n - \mathbf{C}D_{\mathbf{W}+\mathbf{W_o}}^n \left(\mathbf{W}+\mathbf{W}_0\right) X a_n\right\|_2$.
   Using $\|C\|_2 \leq 1$ with high probability, we have $\left\|\mathbf{C}D_{\mathbf{W}+\mathbf{W_o}}^n X a_n\right\|_2 \leq \widetilde{O}(\tau_w \|A\|_1)$

4. Similar to (26), $\left\|V\mathrm{out}^1(X, A)a_n\right\|_2 \leq \tau_v \left\|\mathrm{out}^1(X, A)a_n\right\|_2$ and $\left\langle V_0, \mathrm{out}^1(X, A)a_n\right\rangle_j \sim$
   $\mathcal{N}(0, \left\|\mathrm{out}^1(X, A)a_n\right\|_2^2 \sigma_v^2)$, using Lemma A.2 we can prove it.

5. We write $\mathbf{C}D_{\mathbf{V}+\mathbf{V_o}}^n \mathbf{V}\mathrm{out}^1(X, A)a_n - \mathbf{C}D_{\mathbf{V}+\mathbf{V_o}}^n \left(\mathbf{V}+\mathbf{V}_0\right)\mathrm{out}^1(X, A)a_n =$
   $-\mathbf{C}D_{\mathbf{V_0}}^n \mathbf{V}^{(0)}\mathrm{out}^1(X, A)a_n + \mathbf{C}\left(D_{\mathbf{V_0}}^n - D_{\mathbf{V}+V_0}^n\right)\mathbf{V}_0\mathrm{out}^1(X, A)a_n$. Similar to (27),
   we have $\left\|\mathbf{C}D_{\mathbf{V_0}}^n \mathbf{V}^{(0)}\mathrm{out}^1(X, A)a_n\right\|_2 \leq \widetilde{O}(\sqrt{k}/\sqrt{m}) \cdot O\left(\sigma_v\|\mathrm{out}^1(X, A)a_n\|_2\right)$
   and $\left\|\mathbf{C}\left(D_{\mathbf{V_0}}^n - D_{\mathbf{V}+V_0}^n\right)\mathbf{V}_0\mathrm{out}^1(X, A)a_n\right\|_2 \leq \widetilde{O}\left(\frac{\sqrt{s}}{\sqrt{m}}\tau_v\|\mathrm{out}^1(X, A)a_n\|_2\right)$.
   $\|\mathrm{out}^1(X, A)a_n\|_2 \leq \|\mathrm{out}_n^1(X, A)\|_2\|A\|_1$

6. From 5, it is easy to get.

7. From 3, it is easy to get.

## C.3 EXISTANTIAL

Consider random function $S_n\left((X, A); \mathbf{W}^*\right) = \left(S_n^1\left((X, A); \mathbf{W}^*\right), \ldots, S_n^k\left((X, A); \mathbf{W}^*\right)\right)$ in which

$$S_n^r\left((X, A); \mathbf{W}^*\right) \stackrel{\text{def}}{=} \sum_{i=1}^m a_{r,i} \cdot \left\langle w_i^*, X a_n\right\rangle \cdot \mathbf{1}_{\left\langle w_i^{(0)}, X a_n\right\rangle \geq 0} \tag{34}$$

where $W^*$ is a given matrix, $W^0$ is a random matrix where each $w_i^{(0)}$ is i.i.d. from $\mathcal{N}\left(0, \frac{\mathbf{I}}{m}\right)$ and $a_{r,i}$ is i.i.d. from $\mathcal{N}(0, 1)$.

Based on Lemma B.1. from Li et al. (2022a) and Lemma E.1. from Allen-Zhu & Li (2019), we have

**Lemma C.2** *Given any* $\mathcal{F}$ : $\mathbb{R}^d \rightarrow \mathbb{R}^k$ *with general complexity* $(p, \mathfrak{C}_\mathfrak{s}(\mathcal{F}, \|A\|_1)\|A\|_1, \mathfrak{C}_\varepsilon(\mathcal{F}, \|A\|_1)\|A\|_1)$, *for every* $\epsilon \in \left(0, \frac{1}{pk\mathfrak{C}_\mathfrak{s}(\mathcal{F}, \|A\|_1)\|A\|_1}\right)$, *there exist* $M = \mathrm{poly}\left(\mathfrak{C}_\varepsilon(\mathcal{F}, \|A\|_1), \|A\|_1, 1/\varepsilon\right)$ *such that if* $m \geq M$, *then with high probability there is a construction* $\mathbf{W}^* = (w_1^*, \ldots, w_m^*) \in \mathbb{R}^{m \times d}$ *with*

$$\|\mathbf{W}^*\|_{2,\infty} \leq \frac{kp\mathfrak{C}_\varepsilon(\mathcal{F}, \|A\|_1), \|A\|_1}{m} \quad \text{and} \quad \|\mathbf{W}^*\|_F \leq \widetilde{O}\left(\frac{kp\mathfrak{C}_\mathfrak{s}(\mathcal{F}, \|A\|_1)\|A\|_1}{\sqrt{m}}\right) \tag{35}$$

*satisfying, for every* $x_n \in R^d$ *and* $\|x_n\|_2 \leq 1$*, with probability at least* $1 - e^{-\Omega(\sqrt{m})}$

$$\sum_{r=1}^k |\mathcal{F}_n^r(X, A) - S_n^r(X, A; W^*)| \leq \varepsilon \cdot \|A\|_1 \tag{36}$$

*where* $G_n(X, A; W^*) = \begin{bmatrix} S_n^1(X, A; W^*) \\ \vdots \\ S_n^k(X, A; W^*) \end{bmatrix}$ *and* $S_n(X, A; W^*) = CD_{W+W_0}^n W^* X a_n$

Proof: Define $w_j^* = \sum_{r\in[k]} a_{r,j} \sum_{i\in[p]} a_{r,i}^* h^{(r,i)}\left(\sqrt{m}\left\langle w_j^{(0)}, w_{1,i}^*\right\rangle\right) w_{2,i}^*$ has the same distribution with $\alpha_1$ in Lemma A.6.

Using Lemma A.6 we have $\left|h^{(r,i)}\right| \leq \mathfrak{C}_\varepsilon(\mathcal{F}, \|A\|_1)\|A\|_1$ and using Lemma E.1. from Allen-Zhu & Li (2019), we have for our parameter choice of $m$, with probability at least $1 - e^{-\Omega\left(m\varepsilon^2/\left(k^4p^2\mathfrak{C}_\varepsilon(\mathcal{F}, \|A\|_1)\|A\|_1\right)\right)}$

$$|\mathcal{F}_n^r(X, A) - S_n^r(X, A; W^*)| \leq \frac{\varepsilon}{k}.$$

We have for each $j \in [m]$, with high probability $\left\|w_j^*\right\|_2 \leq \widetilde{O}\left(\frac{kp\mathfrak{C}_\varepsilon(\mathcal{F}, \|A\|_1)\|A\|_1}{m}\right)$. This means $\|\mathbf{W}^*\|_{2,\infty} \leq \widetilde{O}\left(\frac{kp\mathfrak{C}_\varepsilon(\mathcal{F}, \|A\|_1)\|A\|_1}{m}\right)$. As for the Frobenius norm,

$$\|\mathbf{W}^*\|_F^2 = \sum_{j\in[m]} \left\|w_j^*\right\|_2^2 \leq \sum_{j\in[m]} \widetilde{O}\left(\frac{k^2p}{m^2}\right) \cdot \sum_{i\in[p]} h^{(r,i)}\left(\sqrt{m}\left\langle w_j^{(0)}, w_{1,i}^*\right\rangle\right)^2 \tag{37}$$

Applying Hoeffding's concentration, we have with probability at least $1 - e^{-\Omega(\sqrt{m})}$

$$
\begin{aligned}
\sum_{j \in [m]} h^{(r,i)} \left( \sqrt{m} \left\langle w_j^{(0)}, w_{1,i}^* \right\rangle, \sqrt{m} b_j^{(0)} \right)^2 &\leq m \cdot (\mathfrak{C}_{\mathfrak{s}}(\mathcal{F}, \|A\|_1) \|A\|_1^2), \\
&\quad + m^{3/4} \cdot (\mathfrak{C}_{\varepsilon}(\mathcal{F}, \|A\|_1) \|A\|_1)^2, \\
&\leq 2m (\mathfrak{C}_{\mathfrak{s}}(\mathcal{F}, \|A\|_1) \|A\|_1)^2.
\end{aligned}
\tag{38}
$$

Putting this back to (37) we have $\|\mathbf{W}^*\|_F^2 \leq \widetilde{O} \left( \frac{k^2 p^2 (\mathfrak{C}_{\mathfrak{s}}(\mathcal{F}, \|A\|_1) \|A\|_1)^2}{m} \right)$.

**Lemma C.3** *Under the assumptions of Lemma C.1, suppose* $\alpha \in (0,1)$ *and* $\widetilde{\alpha} = \frac{\alpha}{k(p_{\mathcal{F}} \mathfrak{C}_{\mathfrak{s}}(\mathcal{F}, \|A\|_1) + p_{\mathcal{G}} \mathfrak{C}_{\mathfrak{s}}(\mathcal{G}, \|A\|_1))}$, *there exist* $M = \mathrm{poly}\left( \mathfrak{C}_{\widetilde{\alpha}}(\mathcal{F}, \|A\|_1), \mathfrak{C}_{\widetilde{\alpha}}(\mathcal{G}, \|A\|_1), \|A\|_1, \widetilde{\alpha}^{-1} \right)$ *satisfying that for every* $m \geq M, \|\mathbf{W}^*\|_F \leq \widetilde{O}(k p_{\mathcal{F}} \mathfrak{C}_{\mathfrak{s}}(\mathcal{F}))$ *and* $\|\mathbf{V}^*\|_F \leq \widetilde{O}(\widetilde{\alpha} k p_{\mathcal{G}} \mathfrak{C}_{\mathfrak{s}}(\mathcal{G}))$ *with high probability*

1.
$$
\mathbb{E}_{n \in \mathcal{V}, (X, y_n) \sim \mathcal{D}} \left[ \left\| \mathbf{C} D_{\mathbf{W_0}}^n \mathbf{W}^* X a_n - \mathcal{F}_n(X, A) \right\|_2 \right] \leq \widetilde{\alpha}^2 \|A\|_1
\tag{39}
$$

2.
$$
\left\| \mathbf{C} D_{\mathbf{V_0}}^n \mathbf{V}^* \mathrm{out}^1(X, A) a_n - \alpha \mathcal{G}_n \left( \mathrm{out}^1(X, A), A \right) \right\|_2 \leq \widetilde{\alpha}^2 \cdot \left\| \mathrm{out}_n^1(X, A) \right\|_2 \|A\|_1
\tag{40}
$$

3.
$$
\mathbb{E}_{n \in \mathcal{V}, (X, y_n) \sim \mathcal{D}} \left[ \left\| \mathbf{C} D_{\mathbf{W}}^n \mathbf{W}^* X a_n - \mathcal{F}_n(X, A) \right\|_2 \right] \leq O(\widetilde{\alpha}^2 \|A\|_1)
\tag{41}
$$

4.
$$
\begin{aligned}
&\left\| \mathbf{C} D_{\mathbf{V}}^n \mathbf{V}^* \mathrm{out}^1(X, A) a_n - \alpha \mathcal{G}_n \left( \mathrm{out}^1(X, A), A \right) \right\|_2 \\
&\leq \left( \widetilde{\alpha}^2 + O \left( \tau_v \left( \frac{\tau_v}{\sigma_v} \right)^{1/3} \right) \right) \left\| \mathrm{out}_n^1(X, A) \right\|_2 \|A\|_1
\end{aligned}
\tag{42}
$$

5.
$$
\mathbb{E}_{n \in \mathcal{V}, (X, y_n) \sim \mathcal{D}} \left[ \left\| \mathbf{C} D_{\mathbf{W_0 + W}}^n (\mathbf{W}^* - \mathbf{W}) X a_n - (\mathcal{F}_n(X, A) - \mathrm{out}_n^1(X, A)) \right\|_2 \right] \leq \widetilde{\alpha}^2 \|A\|_1
\tag{43}
$$

Proof:

1. Using Lemma C.2, we can find a $\mathbf{W}^*$ satisfying $\left\| \mathbf{C} D_{\mathbf{W_0 + W}}^n \mathbf{W}^* X a_n - \mathcal{F}_n(X, A) \right\|_2$ small enough with probability at least $1 - e^{-\Omega(\sqrt{m})}$.

2. Using Lemma C.2 and $\left\| \mathrm{out}^1(X, A) a_n \right\|_2 \leq \| \mathrm{out}_n^1(X, A) \|_2 \|A\|_1$, we can easily prove it.

3. $\|\mathbf{W}^* X a_n\|_2 \leq O(\|\mathbf{W}^*\|_F \|X a_n\|_2) \leq O(\tau_w \|A\|_1)$. $\left\| \mathbf{C}(D_{\mathbf{W}}^n - D_{\mathbf{W_0}}^n) \mathbf{W}^* X a_n \right\|_2 \leq O(\sqrt{s} \tau_w \|A\|_1 / \sqrt{m})$ where $s$ is the maximum sparsity of $(D_{\mathbf{W}}^n - D_{\mathbf{W_0}}^n)$. Using (26), we know $s \leq O\left( (\frac{\tau_w}{\sigma_w})^{2/3} m^{2/3} \right)$. This, combining with (39) gives

$$
\begin{aligned}
\mathbb{E}_{n \in \mathcal{V}, (X, y_n) \sim \mathcal{D}} \left[ \|\mathbf{C} D_{\mathbf{W}}^n \mathbf{W}^* X a_n - \mathcal{F}_n(X, A)\|_2 \right] &\leq \widetilde{\alpha}^2 \|A\|_1 + O \left( \tau_w (\frac{\tau_w}{\sigma_w})^{1/3} / m^{1/6} \right) \\
&\leq O(\widetilde{\alpha}^2 \|A\|_1)
\end{aligned}
\tag{44}
$$

4. Using (29) and $\|\mathbf{V}^* \mathrm{out}^1(X, A) a_n\|_2 \leq O(\tau_v \| \mathrm{out}_n^1(X, A)\|_2 \|A\|_1)$ we can easily prove it.

5. Using (27) and (41), with larger enough $m$, we can prove it.

C.4 OPTIMIZATION

We write the gradient of loss function as $\nabla_{\mathbf{W}} \mathrm{Obj}_n(\mathbf{W}) = \nabla_{\mathbf{W}} \mathrm{Obj}_n^1(\mathbf{W}) + \nabla_{\mathbf{W}} \mathrm{Obj}_n^2(\mathbf{W})$, where $\nabla_{\mathbf{W}} \mathrm{Obj}_n^1(\mathbf{W}) = \nabla_{\mathbf{W}} \mathrm{out}_n^1(X, A)$ and $\nabla_{\mathbf{W}} \mathrm{Obj}_n^2(\mathbf{W}) = \nabla_{\mathbf{W}} CD_V^n V \mathrm{out}^1(X, A) a_n$, we can write its gradient as follows.

$$
\begin{aligned}
\langle \nabla_{\mathbf{W}} \mathrm{Obj}_n^1(\mathbf{W}), -\mathbf{W}' \rangle &= tr(Xa_n(y_n - \mathrm{out}_n(X, A))^\top CD_{W+W_0}^n \mathbf{W}') \\
&= tr((y_n - \mathrm{out}_n(X, A))^\top CD_{W+W_0}^n W' X a_n) \\
&= \langle y_n - \mathrm{out}_n(X, A), CD_{W+W_0}^n W' X a_n \rangle
\end{aligned}
\tag{45}
$$

$$
\begin{aligned}
\langle \nabla_{\mathbf{W}} \mathrm{Obj}_n^2(\mathbf{W}), -\mathbf{W}' \rangle &= tr(\sum_{i=1}^N a_{ni} X a_i (y_n - \mathrm{out}_n(X, A))^\top CD_{V+V_0}^n (\mathbf{V}^{(0)} + \mathbf{V}) CD_{W+W_0}^i \mathbf{W}') \\
&= tr(\sum_{i=1}^N a_{ni} (y_n - \mathrm{out}_n(X, A))^\top CD_{V+V_0}^n (\mathbf{V}^{(0)} + \mathbf{V}) CD_{W+W_0}^i \mathbf{W}' X a_i) \\
&= tr((y_n - \mathrm{out}_n(X, A))^\top \sum_{i=1}^N a_{ni} CD_{V+V_0}^n (\mathbf{V}^{(0)} + \mathbf{V}) CD_{W+W_0}^i \mathbf{W}' X a_i) \\
&= \langle y_n - \mathrm{out}_n(X, A), CD_{V+V_0}^n (\mathbf{V}^{(0)} + \mathbf{V}) C(D_{W+W_0} \odot W' X A) a_n \rangle
\end{aligned}
\tag{46}
$$

$$
\begin{aligned}
\langle \nabla_{\mathbf{V}} \mathrm{Obj}_n(\mathbf{V}), -\mathbf{V}' \rangle &= tr(\mathrm{out}(X) a_n (y_n - \mathrm{out}_n(X, A))^\top CD_{V+V_0}^n \mathbf{V}') \\
&= tr((y_n - \mathrm{out}_n(X, A))^\top CD_{V+V_0}^n \mathbf{V}' \mathrm{out}(X) a_n) \\
&= \langle y_n - \mathrm{out}_n(X, A), CD_{V+V_0}^n \mathbf{V}' \mathrm{out}(X) a_n \rangle
\end{aligned}
\tag{47}
$$

Let us define $f(\mathbf{W}') = CD_{W+W_0}^n W' X a_n + CD_{V+V_0}^n (\mathbf{V}^{(0)} + \mathbf{V}) C(D_{W+W_0} \odot W' X A) a_n$ and $g(\mathbf{V}') = CD_{V+V_0}^n \mathbf{V}' \mathrm{out}(X) a_n$. Therefore,

$$
\langle \nabla_{\mathbf{W},\mathbf{V}} \mathrm{Obj}_n(\mathbf{W}, \mathbf{V}), (-\mathbf{W}', -\mathbf{V}') \rangle = \langle y_n - \mathrm{out}_n(X, A), f(\mathbf{W}') + g(\mathbf{V}') \rangle
\tag{48}
$$

**Claim C.4** *We have that for all* $\mathbf{W}$ *and* $\mathbf{V}$ *satisfying* $\|\mathbf{W}\|_F \le \tau_w$ *and* $\|\mathbf{V}\|_F \le \tau_v$, *it holds that*

$$
\|\nabla_{\mathbf{W}} \mathrm{Obj}(\mathbf{W}, \mathbf{V}; (x, y))\|_F \le \|A\|_1 \|y_n - \mathrm{out}_n(X, A)\|_2 \cdot O(\sigma_v + 1)
\tag{49}
$$

$$
\|\nabla_{\mathbf{V}} \mathrm{Obj}(\mathbf{W}, \mathbf{V}; (x, y))\|_F \le \tau_w \|A\|_1 \|y_n - \mathrm{out}_n(X, A)\|_2 \cdot O(1)
\tag{50}
$$

Proof:

$$
\begin{aligned}
\|\nabla_{\mathbf{W}} \mathrm{Obj}(\mathbf{W}, \mathbf{V}; (x, y))\|_F &= \|Xa_n(y_n - \mathrm{out}_n(X, A))^\top \\
&\quad \times (CD_{W+W_0}^n + CD_{V+V_0}^n (\mathbf{V}^{(0)} + \mathbf{V}) CD_{W+W_0}^n)\|_F \\
&\le \|Xa_n\|_2 \|y_n - \mathrm{out}_n(X, A)\|_2 \\
&\quad \times \|CD_{W+W_0}^n + CD_{V+V_0}^n (\mathbf{V}^{(0)} + \mathbf{V}) CD_{W+W_0}^n\|_2 \\
&\le \|A\|_1 \|y_n - \mathrm{out}_n(X, A)\|_2 \cdot O(\sigma_v + 1)
\end{aligned}
\tag{51}
$$

In (51), the last inequality uses $\|V^{(0)}\|_2 = O(\tau_v)$ and $\|C\|_2 \le 1$.

$$
\begin{aligned}
\|\nabla_{\mathbf{V}} \mathrm{Obj}(\mathbf{W}, \mathbf{V}; (x, y))\|_F &= \|\mathrm{out}_n(X, A) a_n (y_n - \mathrm{out}_n(X, A))^\top CD_{V+V_0}^n\|_F \\
&\le \|\mathrm{out}_n(X, A) a_n\|_2 \|y_n - \mathrm{out}_n(X, A)\|_2 \|CD_{V+V_0}^n\|_2 \\
&\le \tau_w \|A\|_1 \|y_n - \mathrm{out}_n(X, A)\|_2 \cdot O(1)
\end{aligned}
\tag{52}
$$

In (52), the last inequality uses (28) and $\|C\|_2 \le 1$.

**Claim C.5** *In the setting of Lemma C.1, we have* $f\left(\mathbf{W}^* - \mathbf{W}\right) + g\left(\mathbf{V}^* - \mathbf{V}\right) = \mathcal{H}_{n,A^*}(X,A) - \mathrm{out}_n(X,A) + Err_n$ *with*

$$\underset{n\in\mathcal{V},(X,y_n)\sim\mathcal{D}}{\mathbb{E}} \|Err_n\|_2 \leq \underset{n\in\mathcal{V},(X,y_n)\sim\mathcal{D}}{\mathbb{E}} \left[ O\left(\tau_v \|A\|_1 + \alpha\mathfrak{L}_\mathcal{G}\|A\|_1\right) \cdot \|\mathcal{H}_{n,A^*}(X,A) - \mathrm{out}_n(X,A)\|_2 \right]$$
$$+ O\left(\tau_v^2 \|A\|_1^2 \mathfrak{B}_\mathcal{F} + \tau_v\widetilde{\alpha}^2 \|A\|_1^2 + \alpha\tau_v \|A\|_1^2 \mathfrak{L}_\mathcal{G}\mathfrak{B}_\mathcal{F}\right) \tag{53}$$

Proof: Based on the definition of $f(\mathbf{W}')$ and $g(\mathbf{V}')$, we have

$$
\begin{aligned}
f\left(\mathbf{W}^* - \mathbf{W}; X, a_n^*\right) + g\left(\mathbf{V}^* - \mathbf{V}; X, a_n^*\right) = {} & CD_{W+W_0}^n (W^* - W) X a_n \\
& + CD_{V+V_0}^n (V^* - V)\,\mathrm{out}(X) a_n \\
& + CD_{V+V_0}^n (\mathbf{V}^{(0)} + \mathbf{V}) C(D_{W+W_0} \odot (W^* - W) X A) a_n \\
= {} & \underbrace{CD_{V+V_0}^n (\mathbf{V}^{(0)} + \mathbf{V}) C(D_{W+W_0} \odot (W^* - W) X A) a_n}_{\clubsuit} \\
& + \underbrace{CD_{W+W_0}^n W^* X a_n + CD_{V+V_0}^n V^*\,\mathrm{out}^1(X,A) a_n}_{\spadesuit} \\
& - \underbrace{(CD_{W+W_0}^n W X a_n + CD_{V+V_0}^n V\,\mathrm{out}^1(X,A) a_n)}_{\diamondsuit}
\end{aligned}
\tag{54}
$$

1. For the $\clubsuit$ term,

$$
\begin{aligned}
\clubsuit &\leq \left( \left\| CD_{V+V_0}^n \mathbf{V}^{(0)} \right\|_2 + \|\mathbf{C}\|_2^2 \|\mathbf{V}\|_2 \right) \|C(D_{W+W_0} \odot (W^* - W) X A) a_n\|_2 \\
&\leq O(1) \cdot O\left(\tau_v\right) \cdot \sum_{i=1}^N a_{ni} \left( \|\mathcal{F}(x) - \mathrm{out}_n^1(X,A)\|_2 + O\left(\widetilde{\alpha}^2 \|A\|_1\right) \right) \\
&\leq O\left(\tau_v\right) \left( \|\mathcal{F}(x) - \mathrm{out}_i(x)\|_2 \|A\|_1 + O\left(\widetilde{\alpha}^2 \|A\|_1^2\right) \right)
\end{aligned}
\tag{55}
$$

together with $\tau_v \leq \frac{1}{\mathrm{polylog}\,(m)}\sigma_v$.

2. For the $\spadesuit$ term,

$$
\begin{aligned}
\spadesuit - \left(\mathcal{F}_n(X,A) + \alpha\mathcal{G}(\mathcal{F}(x), a_n)\right) = {} & CD_{W+W_0}^n W^* X a_n - \mathcal{F}_n(X,A) \\
& + CD_{V+V_0}^n V^*\,\mathrm{out}^1(X,A) a_n - \alpha\mathcal{G}\left(\mathrm{out}_n^1(X,A), a_n\right) \\
& + \alpha\mathcal{G}\left(\mathrm{out}_n^1(X,A), a_n\right) - \alpha\mathcal{G}\left(\mathcal{F}(x), a_n\right)
\end{aligned}
\tag{56}
$$

The first term uses (39), the second term uses (40) and the third term uses the Lipscthiz continuity of $\mathcal{G}$, so we have

$$
\begin{aligned}
\|\spadesuit - \left(\mathcal{F}(x) + \alpha\mathcal{G}(\mathcal{F}(x))\right)\|_2 \leq {} & O\left(\widetilde{\alpha}^2 + \tau_v(\frac{\tau_v}{\sigma_v})^{1/3}\right) \cdot \|\mathrm{out}_n^1(X,a_n)\|_2 \|A\|_1 \\
& + O\left(\alpha\mathfrak{L}_\mathcal{G}\right) \|\mathcal{F}(X) a_n - \mathrm{out}_n^1(X) a_n\|_2 \\
\leq {} & O\left(\tau_v^2\right) \cdot \|\mathrm{out}_n^1(x)\|_2 \|A\|_1 + O\left(\alpha\mathfrak{L}_\mathcal{G}\right) \|\mathcal{F}_n(x) - \mathrm{out}_n^1(x)\|_2 \|A\|_1
\end{aligned}
\tag{57}
$$

We use $\frac{1}{\sigma_v} \leq \tau_v^2$ and definition of $\widetilde{\alpha}$.

3. For the $\diamondsuit$ term,

$$\|\diamondsuit - \mathrm{out}_n(X)\|_2 \leq O\left(\left(\|\mathrm{out}_n^1(x)\|_2 \|A\|_1\right) \tau_v^2\right) \tag{58}$$

where the inequality uses (27) and (30).

In sum, we have

$$Err \overset{\text{def}}{=} f\left(\mathbf{W}^* - \mathbf{W}; x\right) + g\left(\mathbf{V}^* - \mathbf{V}; x\right) - \left(\mathcal{F}(x) + \alpha\mathcal{G}(\mathcal{F}(x)\right) - \mathrm{out}_n(X,A) \tag{59}$$

satisfy

$$
\begin{aligned}
\mathbb{E}_{n \in \mathcal{V}, (X, y_n) \sim \mathcal{D}} \|Err\|_2 \leq \mathbb{E}_{n \in \mathcal{V}, (X, y_n) \sim \mathcal{D}} \Big[ & O\left(\tau_v \|A\|_1 + \alpha \mathfrak{L}_{\mathcal{G}} \|A\|_1\right) \\
& \times \|\mathcal{F}(x) - \text{out}_n^1(X, A)\|_2 + O\left(\|\text{out}_n^1(x)\|_2 \|A\|_1\right) \tau_v^2 \Big] \\
& + O\left(\tau_v \widetilde{\alpha}^2 \|A\|_1^2\right)
\end{aligned}
\tag{60}
$$

Using $\|\text{out}_n^1(x)\|_2 \leq \|\text{out}_n^1(X, A) - \mathcal{F}(x)\|_2 + \mathfrak{B}_{\mathcal{F}}$, we have

$$
\begin{aligned}
\mathbb{E}_{n \in \mathcal{V}, (X, y_n) \sim \mathcal{D}} \|Err\|_2 \leq \mathbb{E}_{n \in \mathcal{V}, (X, y_n) \sim \mathcal{D}} & \left[O\left(\tau_v \|A\|_1 + \alpha \mathfrak{L}_{\mathcal{G}} \|A\|_1\right) \cdot \|\mathcal{H}(x) - \text{out}_n(X, A)\|_2\right] \\
& + O\left(\tau_v^2 \|A\|_1 \mathfrak{B}_{\mathcal{F}} + \tau_v \widetilde{\alpha}^2 \|A\|_1^2\right) \\
& + O\left(\tau_v \|A\|_1 + \alpha \mathfrak{L}_{\mathcal{G}} \|A\|_1\right) \cdot \left(\tau_v \|A\|_1 \mathfrak{B}_{\mathcal{F}} + \alpha \mathfrak{B}_{\mathcal{F} \circ \mathcal{G}}\right)
\end{aligned}
\tag{61}
$$

Using $\mathfrak{B}_{\mathcal{F} \circ \mathcal{G}} \leq \sqrt{k} p_{\mathcal{G}} \mathfrak{C}_{\mathfrak{s}}(\mathcal{G}, \|A\|_1 \mathfrak{B}_{\mathcal{F}})(\|A\|_1 \mathfrak{B}_{\mathcal{F}})^2 \mathfrak{B}_{\mathcal{F}} \leq \frac{\tau_v}{\alpha} \mathfrak{B}_{\mathcal{F}}$, so we have

$$
\begin{aligned}
\mathbb{E}_{n \in \mathcal{V}, (X, y_n) \sim \mathcal{D}} \|Err\|_2 \leq \mathbb{E}_{n \in \mathcal{V}, (X, y_n) \sim \mathcal{D}} & \left[O\left(\tau_v \|A\|_1 + \alpha \mathfrak{L}_{\mathcal{G}} \|A\|_1\right) \cdot \|\mathcal{H}(x) - \text{out}_n(X, A)\|_2\right] \\
& + O\left(\tau_v^2 \|A\|_1^2 \mathfrak{B}_{\mathcal{F}} + \tau_v \widetilde{\alpha}^2 \|A\|_1^2 + \alpha \tau_v \|A\|_1^2 \mathfrak{L}_{\mathcal{G}} \mathfrak{B}_{\mathcal{F}}\right)
\end{aligned}
\tag{62}
$$

**Claim C.6** *In the setting of Lemma C.1, if $\tau_v \|A\|_1 \leq \frac{1}{\text{polylog}(m)}$, we have*

$$
\left\|\text{out}_n^1(X, A) - \mathcal{F}_n(X)\right\|_2 \leq 2 \left\|\text{out}_n^1(X, A) - \mathcal{H}_n(X, A)\right\|_2 + \alpha \mathfrak{B}_{\mathcal{F} \circ \mathcal{G}} + \widetilde{O}\left(\tau_v \|A\|_1 \mathfrak{B}_{\mathcal{F}}\right)
\tag{63}
$$

Proof: Using 32 and $\|\mathcal{G}(\mathcal{F}(X), a_n)\|_2 \leq \mathfrak{B}_{\mathcal{F} \circ \mathcal{G}}$, we have

$$
\begin{aligned}
\left\|\text{out}_n^1(X, A) - \mathcal{F}_n(X)\right\|_2 \leq & \left\|\text{out}_n^1(X, A) - \mathcal{H}_n(X, A)\right\|_2 + \alpha \mathfrak{B}_{\mathcal{F} \circ \mathcal{G}} \\
& + \widetilde{O}\left(\tau_v \|A\|_1 \left(\|\text{out}_n^1(X, a_n) - \mathcal{F}_n(X, A)\|_2 + \mathfrak{B}_{\mathcal{F}}\right)\right)
\end{aligned}
\tag{64}
$$

Using $\tau_v \|A\|_1$ small enough, we have

$$
\left\|\text{out}_n^1(X, A) - \mathcal{F}_n(X)\right\|_2 \leq 2 \left\|\text{out}_n^1(X, A) - \mathcal{H}_n(X, A)\right\|_2 + \alpha \mathfrak{B}_{\mathcal{F} \circ \mathcal{G}} + \widetilde{O}\left(\tau_v \|A\|_1 \mathfrak{B}_{\mathcal{F}}\right)
\tag{65}
$$

## C.5 PROOF OF THEOREM 3.2

Proof: Using (48) and Claim C.5, in iteration $t$, we have

$$
\begin{aligned}
& \langle \nabla_{\mathbf{W}, \mathbf{v}} \text{Obj}_n\left(\mathbf{W}_t, \mathbf{V}_t\right), \left(\mathbf{W}_t - \mathbf{W}^*, \mathbf{V}_t - \mathbf{V}^*\right)\rangle \\
& = \langle y_n - \text{out}\left(\mathbf{W}_t, \mathbf{V}_t\right), f\left(\mathbf{W}^* - \mathbf{W}\right) + g\left(\mathbf{V}^* - \mathbf{V}\right)\rangle \\
& = \langle y_n - \text{out}_n\left(\mathbf{W}_t, \mathbf{V}_t\right), \mathcal{H}_{n, A^*}(X) - \text{out}_n\left(\mathbf{W}_t, \mathbf{V}_t\right) + Err_t\rangle
\end{aligned}
\tag{66}
$$

We also have

$$
\begin{aligned}
\|W_{t+1} - W^*\|_F^2 & = \|W_t - \eta_w \nabla_W \text{Obj}_n(W_t) - W^*\|_F^2 \\
& = \|W_t - W^*\|_F^2 - 2\eta_w \langle \nabla_W \text{Obj}_n(W_t), W_t - W^*\rangle \\
& \quad + \eta_w^2 \|\nabla_W \text{Obj}_n(W_t)\|_F^2,
\end{aligned}
\tag{67}
$$

$$
\begin{aligned}
\|V_{t+1} - V^*\|_F^2 & = \|V_t - \eta_v \nabla_V \text{Obj}_n(V_t) - V^*\|_F^2 \\
& = \|V_t - V^*\|_F^2 - 2\eta_v \langle \nabla_V \text{Obj}_n(V_t), V_t - V^*\rangle \\
& \quad + \eta_v^2 \|\nabla_V \text{Obj}_n(V_t)\|_F^2
\end{aligned}
\tag{68}
$$

Using Algorithm 1, we have $\mathbf{W}_{t+1} = \mathbf{W}_t - \eta_w \nabla_{\mathbf{W}} \mathrm{Obj}_n(\mathbf{W}_t, \mathbf{V}_t)$ and $\mathbf{V}_{t+1} = \mathbf{V}_t - \eta_v \nabla_{\mathbf{V}} \mathrm{Obj}_n(\mathbf{W}_t, \mathbf{V}_t)$, so we have

$$
\begin{aligned}
&\langle \nabla_{\mathbf{W},\mathbf{V}} \mathrm{Obj}_t(\mathbf{W}_t, \mathbf{V}_t), (\mathbf{W} - \mathbf{W}^*, \mathbf{V} - \mathbf{V}^*)) \rangle \\
&= \underbrace{\frac{\eta_w}{2} \| \nabla_{\mathbf{W}} \mathrm{Obj}_t(\mathbf{W}_t, \mathbf{V}_t) \|_F^2 + \frac{\eta_v}{2} \| \nabla_{\mathbf{V}} \mathrm{Obj}_t(\mathbf{W}_t, \mathbf{V}_t) \|_F^2}_{\heartsuit} \\
&\quad + \frac{1}{2\eta_w} \| \mathbf{W}_t - \mathbf{W}^* \|_F^2 - \frac{1}{2\eta_w} \| \mathbf{W}_{t+1} - \mathbf{W}^* \|_F^2 + \frac{1}{2\eta_v} \| \mathbf{V}_t - \mathbf{V}^* \|_F^2 - \frac{1}{2\eta_v} \| \mathbf{V}_{t+1} - \mathbf{V}^* \|_F^2
\end{aligned}
\tag{69}
$$

Using Claim C.4 and change all $A$ to $A^*$, we have

$$
\begin{aligned}
\heartsuit &\le O\left(\eta_w \sigma_v^2 + \eta_v \tau_w^2\right) \cdot \|A\|_1^2 \| y_n - \mathrm{out}_n(X, A^*) \|_2^2 \\
&\le O\left(\eta_w \sigma_v^2 + \eta_v \tau_w^2\right) \cdot \|A\|_1^2 \left( \| \mathcal{H}_{n,A^*}(X) - \mathrm{out}_n(X, A^*) \|_2^2 + \| \mathcal{H}_{n,A^*}(X) - y_n \|_2^2 \right)
\end{aligned}
\tag{70}
$$

Therefore, as long as $O\left(\eta_w \sigma_v^2 + \eta_v \tau_w^2\right) \le 0.1$, it satisfies

$$
\begin{aligned}
\frac{1}{4} \| \mathcal{H}_{n,A^*}(X) - \mathrm{out}_n(X, A^*) \|_2^2 \le{}& 2 \| \mathrm{Err}_t \|_2^2 + 4 \| \mathcal{H}_{n,A^*}(X) - y_n \|_2^2 \\
&+ \frac{1}{2\eta_w} \| \mathbf{W}_t - \mathbf{W}^* \|_F^2 - \frac{1}{2\eta_w} \| \mathbf{W}_{t+1} - \mathbf{W}^* \|_F^2 \\
&+ \frac{1}{2\eta_v} \| \mathbf{V}_t - \mathbf{V}^* \|_F^2 - \frac{1}{2\eta_v} \| \mathbf{V}_{t+1} - \mathbf{V}^* \|_F^2
\end{aligned}
\tag{71}
$$

After telescoping for $t = 0, 1, \ldots, T_0 - 1$,

$$
\begin{aligned}
&\frac{\| \mathbf{W}_{T_0} - \mathbf{W}^* \|_F^2}{2\eta_w T_0} + \frac{\| \mathbf{W}_{T_0} - \mathbf{V}^* \|_F^2}{2\eta_v T_0} + \frac{1}{2T_0} \sum_{t=0}^{T_0 - 1} \| \mathcal{H}_{n,A^*}(X) - \mathrm{out}_n(X, A^*) \|_2^2 \\
&\le \frac{\| \mathbf{W}^* \|_F^2}{2\eta_w T_0} + \frac{\| \mathbf{V}^* \|_F^2}{2\eta_v T_0} + \frac{O(1)}{T_0} \sum_{t=0}^{T_0 - 1} \| Err_t \|_2^2 + \| \mathcal{H}_{n,A^*}(X) - y_t \|_2^2.
\end{aligned}
\tag{72}
$$

Using $O\left(\tau_v \|A\|_1 + \alpha \mathfrak{L}_{\mathcal{G}}\right) \le 0.1$, we have

$$
\frac{1}{4T} \sum_{t=0}^{T-1} \mathbb{E}_{n \in \mathcal{V}, (X, y_n) \sim \mathcal{Z}} \| \mathcal{H}_{n,A^*}(X) - \mathrm{out}_n(X, A^*) \|_2^2 \le \frac{\| \mathbf{W}^* \|_F^2}{2\eta_w T} + \frac{\| \mathbf{V}^* \|_F^2}{2\eta_v T} + O\left(\mathrm{OPT} + \epsilon_0\right)
\tag{73}
$$

where

$$
\begin{aligned}
\epsilon_0 &= \Theta\left( \widetilde{\alpha}^2 \tau_v \|A^*\|_1^2 + \tau_v^2 \|A^*\|_1 \mathfrak{B}_{\mathcal{F}} + \alpha \tau_v \|A^*\|_1 \mathfrak{L}_{\mathcal{G}} \mathfrak{B}_{\mathcal{F}} \right)^2 \\
&= \widetilde{\Theta}\left( \widetilde{\alpha}^2 \tau_v \|A^*\|_1^2 + \alpha^2 (k p_{\mathcal{G}} \mathfrak{C}_{\mathfrak{s}}(\mathcal{G}, \mathfrak{B}_{\mathcal{F}} \|A^*\|_1)^2 (\mathfrak{B}_{\mathcal{F}} \|A^*\|_1)^3 \right. \\
&\qquad \left. + \alpha^2 k p_{\mathcal{G}} \mathfrak{C}_{\mathfrak{s}}(\mathcal{G}, \mathfrak{B}_{\mathcal{F}} \|A^*\|_1)(\mathfrak{B}_{\mathcal{F}} \|A^*\|_1)^3 \|A^*\|_1 \right)^2 \\
&= \widetilde{\Theta}\left( \alpha^4 \left(p_{\mathcal{G}} \mathfrak{C}_{\mathfrak{s}}(\mathcal{G}, \mathfrak{B}_{\mathcal{F}} \|A^*\|_1)\right)^4 \left(\|A^*\|_1 \mathfrak{B}_{\mathcal{F}}\right)^6 \right)
\end{aligned}
\tag{74}
$$

In practice, for computational efficiency, we use the sampled adjacency matrix $A^t$ in the learning network, so we should consider the discrepancy between the target function and the practical output

$$
\begin{aligned}
\left\| \mathcal{H}_{n,A^*}(X) - \mathrm{out}_n\left(X, A^{1t}, A^{2t}\right) \right\|_2^2 \le{}& \left\| \mathcal{H}_{n,A^*}(X) - \mathrm{out}_n(X, A^*) \right\|_2^2 \\
&+ \left\| \mathrm{out}_n\left(X, A^{1t}, A^{2t}\right) - \mathrm{out}_n(X, A^*) \right\|_2^2
\end{aligned}
\tag{75}
$$

We have already considered $\| \mathcal{H}_{n,A^*}(X) - \mathrm{out}_n(X, A^*) \|_2^2$ and

$$
\begin{aligned}
\left\| \mathrm{out}_n\left(X, A^{1t}, A^{2t}\right) - \mathrm{out}_n(X, A^*) \right\|_2 \le{}& \left\| C\sigma(WXa_n^{1t}) - C\sigma W X a_n^* \right\|_2 \\
&+ \left\| C\sigma(V \mathrm{out}_n^1(XA^{1t})a_n^{2t}) - C\sigma(V \mathrm{out}_n^1(XA^*)a_n^*) \right\|_2
\end{aligned}
\tag{76}
$$

Using (28), we have

$$\left\|C\sigma(WXa_n^{1t}) - C\sigma WXa_n^*\right\|_2 \leq \tau_w \left\|Xa_n^{1t} - Xa_n^*\right\|_2 \leq \left\|\mathrm{Err}_t\right\|_2 \tag{77}$$

For the above equation to hold, it requires

$$\left\|A^{1t} - A^*\right\|_1 \leq \left\|\frac{Err_t}{\tau_w}\right\|_2 \tag{78}$$

Using $\left\|A^*\right\|_1 \leq O(1)$ and (32), we have

$$\left\|C\sigma(V\,\mathrm{out}_n^1(XA^{1t})a_n^{2t}) - C\sigma(V\,\mathrm{out}_n^1(XA^*)a_n^*)\right\|_2 \leq \tau_v \left\|\mathrm{out}_n^1(XA^{1t})a_n^{2t} - \mathrm{out}_n^1(XA^*)a_n^*\right\|_2$$
$$\leq \tau_v\tau_w \left\|A^{2t} - A^*\right\|_1 \leq \left\|\mathrm{Err}_t\right\|_2 \tag{79}$$

For the above equation to hold, it requires $\left\|A^{2t} - A^*\right\|_1 \leq \left\|\frac{Err_t}{\tau_v\tau_w}\right\|_2$.

Under assumptions of Lemma C.7, with high probability, we can ensure $\left\|A^{1t} - A^*\right\|_2 \leq \left\|\frac{Err_t}{\tau_w}\right\|_2$, $\left\|A^{2t} - A^*\right\|_1 \leq \left\|\frac{Err_t}{\tau_v\tau_w}\right\|_2$.

Using $\left\|\mathbf{W}^*\right\|_F \leq \tau_w/10, \left\|\mathbf{V}^*\right\|_F \leq \tau_v/10$ and $\epsilon \geq \mathrm{OPT} + \epsilon_0$, we have

$$\frac{1}{T}\sum_{t=0}^{T-1} \mathbb{E}_{n\in\mathcal{V},(X,y_n)\sim\mathcal{D}} \left\|\mathcal{H}_{n,A^*}(X) - \mathrm{out}_n\left(X, A^{1t}, A^{2t}\right)\right\|_2^2 \leq O(\epsilon) \tag{80}$$

as long as $T \geq \Omega\left(\frac{\tau_w^2/\eta_w + \tau_v^2/\eta_v}{\epsilon}\right)$.

Finally, we should check $\left\|\mathbf{W}_t\right\|_F \leq \tau_w$ and $\left\|\mathbf{V}_t\right\|_F \leq \tau_v$ hold.

$$\frac{\left\|\mathbf{W}_{T_0} - \mathbf{W}^*\right\|_F^2}{2\eta_w T_0} + \frac{\left\|\mathbf{V}_{T_0} - \mathbf{V}^*\right\|_F^2}{2\eta_v T_0} \leq \frac{\left\|\mathbf{W}^*\right\|_F^2}{2\eta_w T_0} + \frac{\left\|\mathbf{V}^*\right\|_F^2}{2\eta_v T_0} + O(\epsilon) + \widetilde{O}\left(\frac{\tau_w\|A^*\|_1}{\sqrt{T_0}}\right) \tag{81}$$

Using the relationship $\frac{\tau_w^2}{\eta_w} = \frac{\tau_v^2}{\eta_v}$, we have

$$\frac{\left\|\mathbf{W}_{T_0}\right\|_F^2}{\tau_w^2} + \frac{\left\|\mathbf{V}_{T_0}\right\|_F^2}{\tau_v^2} \leq \frac{4\left\|\mathbf{W}^*\right\|_F^2}{\tau_w^2} + \frac{4\left\|\mathbf{V}^*\right\|_F^2}{\tau_v^2} + 0.1 + \widetilde{O}\left(\frac{\eta_w\|A^*\|_1\sqrt{T_0}}{\tau_w}\right) \tag{82}$$

Therefore, choosing $T = \widetilde{\Theta}\left(\frac{\tau_w^2}{\|A^*\|_1\min\{1,\epsilon^2\}}\right)$ and $\eta_w = \widetilde{\Theta}(\min\{1,\epsilon\}) \leq 0.1$, we can ensure $\frac{\left\|\mathbf{W}_{T_0}\right\|_F^2}{\tau_w^2} + \frac{\left\|\mathbf{V}_{T_0}\right\|_F^2}{\tau_v^2} \leq 1$.

## C.6 GRAPH SAMPLING

**Lemma C.7** *Given a graph $G$ with the minimum degree $\delta(G) \geq \Omega(\left\|\frac{\tau_w}{Err_t}\right\|_2)$, in iteration $t$, for the first layer $A^{1t}$ is generated from the sampling strategy with the sampling probability $p_{ij}^1 \leq O(\frac{\sqrt{d_id_j}\|Err_t\|_2}{n_{ij}\tau_w})$ and for the second layer $A^{2t}$ is generated from the sampling strategy with the sampling probability $p_{ij}^2 \leq O(\frac{\sqrt{d_id_j}\|Err_t\|_2}{n_{ij}\tau_w\tau_v})$, we have*

$$\Pr\left[\left\|A^{1t} - A^*\right\|_1 \leq O(\left\|\frac{Err_t}{\tau_w}\right\|_2)\right] \leq e^{-\Omega(\|Err_t\|_2\sqrt{d_id_j}/\tau_w)} \tag{83}$$

$$\Pr\left[\left\|A^{2t} - A^*\right\|_1 \leq O(\left\|\frac{Err_t}{\tau_w\tau_v}\right\|_2)\right] \leq e^{-\Omega(\|Err_t\|_2\sqrt{d_id_j}/\tau_w\tau_v)} \tag{84}$$

Proof: The difference between $A_{B_{ij}}^t$ and $A_{B_{ij}}^*$ is

$$\Delta_{B_{ij}} = \left\| A_{B_{ij}}^t - A_{B_{ij}}^* \right\| = \sum_{i=1}^{n_{ij}} (a_{ij}^t - a_{ij}^*) \tag{85}$$

where $n_{ij}$ is the number of elements in $A_{B_{ij}}^*$ and $(a_{ij}^t - a_{ij}^*)$ are iid, with $\mu_{ij} = \mathbb{E}[\Delta_{B_{ij}}] = n_{ij} p_{ij} \frac{1}{\sqrt{d_i d_j}}$. The Moment-generating function of $(a_{ij}^t - a_{ij}^*)$ is

$$
\begin{aligned}
M_{(a_{ij}^t - a_{ij}^*)}(s) &= \mathbb{E}\left[ e^{s(a_{ij}^t - a_{ij}^*)} \right] \\
&= e^{s \frac{1}{\sqrt{d_i d_j}}} p_{ij} + e^{s \cdot 0}(1 - p_{ij}) \\
&= 1 + p_{ij} \left( e^{\frac{s}{\sqrt{d_i d_j}}} - 1 \right) \\
&\leq exp\left( e^{\frac{s}{\sqrt{d_i d_j}}} - 1 \right)
\end{aligned} \tag{86}
$$

Thus, for any $t > 0$, using Markov's inequality and the definition of MGF, we have

$$
\begin{aligned}
\mathbb{P}(\Delta_{B_{ij}} \geq k) &\leq \min_{s>0} \frac{\prod_{i=1}^{n_{ij}} M_{(a_{ij}^t - a_{ij}^*)}(s)}{e^{tk}} \\
&= \min_{t>0} \frac{e^{\mu \sqrt{d_i d_j} \left( e^{\frac{s}{\sqrt{d_i d_j}}} - 1 \right)}}{e^{tk}}
\end{aligned} \tag{87}
$$

If $0 \leq \delta_{ij} \leq 1$, we plug in $k_{ij} = (1 + \delta_{ij})\mu_{ij}$ and the optimal value of $s_{ij} = \sqrt{d_i d_j} \ln(1 + \epsilon_{ij})$ to the above equation:

$$\mathbb{P}(\Delta_{B_{ij}} \geq (1 + \delta_{ij})\mu_{ij}) \leq \left( \frac{e^{\epsilon_{ij}}}{(1 + \epsilon_{ij})^{(1+\epsilon_{ij})}} \right)^{\mu_{ij}\sqrt{d_i d_j}} \leq \exp\left( \frac{-\epsilon_{ij}^2 \mu_{ij} \sqrt{d_i d_j}}{3} \right) \tag{88}$$

$$
\begin{aligned}
(1 + \delta_{ij})^{(1+\delta_{ij})} &= exp[(1 + \delta_{ij}) ln(1 + \delta_{ij})] \\
&= exp(\delta_{ij} + \frac{\delta_{ij}^2}{2} - \frac{\delta_{ij}^3}{6} + o(\delta_{ij}^4)) \geq exp(\delta_{ij} + \frac{\delta_{ij}^2}{2} - \frac{\delta_{ij}^3}{6})
\end{aligned} \tag{89}
$$

Let $\delta_{ij} = 1$, $\mu_{ij} = \Theta(Err_t)$, and $d_i \geq \Omega(\frac{1}{Err_t})$, we have $p_{ij} \leq O(\frac{\sqrt{d_i d_j} Err_t}{n_{ij}})$.

## C.7 Sample Complexity

**Lemma C.8** *Given a graph $G$ with $|V(G)| = N$, if the maximum degree $\Delta(G) \leq O((N\epsilon^2)^{\frac{1}{4}})$ and sample complexity $\Omega \geq O(\frac{\Delta(G)^2 (\tau_w \|A\|_1)^2 \log N}{\epsilon^2})$, with probability $1 - N^{-\tau_w \|A\|_1}$, we have $\left| \underset{n \in \mathcal{V}, (X, y_n) \sim \mathcal{Z}}{\mathbb{E}} \left\| y_n - \text{out}_n \left( X, A^{1t}, A^{2t} \right) \right\|_2 - \underset{n \in \mathcal{V}, (X, y_n) \sim \mathcal{D}}{\mathbb{E}} \left\| y_n - \text{out}_n \left( X, A^{1t}, A^{2t} \right) \right\|_2 \right| \leq \epsilon.$*

Proof: For the set of samples Z define

$$\underset{n \in \mathcal{V}, (X, y_n) \sim \mathcal{Z}}{\mathbb{E}} \left\| y_n - \text{out}_n \left( X, A^{1t}, A^{2t} \right) \right\|_2 = \frac{1}{\Omega} \sum_{n=1}^{\Omega} \left\| y_n - \text{out}_n \left( X, A^{1t}, A^{2t} \right) \right\|_2 \tag{90}$$

Denote the generalization error as

$$
\begin{aligned}
&\left| \underset{n \in \mathcal{V}, (X, y_n) \sim \mathcal{Z}}{\mathbb{E}} \left\| y_n - \text{out}_n \left( X, A^{1t}, A^{2t} \right) \right\|_2 - \underset{n \in \mathcal{V}, (X, y_n) \sim \mathcal{D}}{\mathbb{E}} \left\| y_n - \text{out}_n \left( X, A^{1t}, A^{2t} \right) \right\|_2 \right| \\
&= \left| \underset{n \in \mathcal{V}, (X, y_n) \sim \mathcal{Z}}{\mathbb{E}} \left\| \text{out}_n \left( X, A^{1t}, A^{2t} \right) \right\|_2 - \underset{n \in \mathcal{V}, (X, y_n) \sim \mathcal{D}}{\mathbb{E}} \left\| \text{out}_n \left( X, A^{1t}, A^{2t} \right) \right\|_2 \right|
\end{aligned}
$$

By Hoeffding's inequality and $\left\| \text{out}_n(X, a_n) \right\|_2 \le O(\tau_w \left\| A \right\|_1)$, we have

$$\mathbb{E}\left[ e^{s \left| \underset{n \in \mathcal{V}, (X, y_n) \sim \mathcal{Z}}{\mathbb{E}} \left\| \text{out}_n \left( X, A^{1t}, A^{2t} \right) \right\|_2 - \underset{n \in \mathcal{V}, (X, y_n) \sim \mathcal{D}}{\mathbb{E}} \left\| \text{out}_n \left( X, A^{1t}, A^{2t} \right) \right\|_2 \right|} \right] \le e^{\frac{(s\tau_w \|A\|_1)^2}{8}} \tag{91}$$

Define maximum degree of G is $\Delta(G)$. It is easy to know that $\left\| \text{out}_n \left( X, A^{1t}, A^{2t} \right) \right\|_2$ is dependent with at most its second order neighbor, so the maximum number of nodes related with $\left\| \text{out}_n \left( X, A^{1t}, A^{2t} \right) \right\|_2$ is $\Delta(G)^2$. By Lemma 7 in Shuai, we have

$$\mathbb{E} e^{s \sum_{n=1}^{\Omega} \left\| \text{out}_n \left( X, A^{1t}, A^{2t} \right) \right\|_2} \le e^{\Delta(G)^2 (s\tau_w \|A\|_1)^2 \Omega/8} \tag{92}$$

$$\mathbb{P}\left( \left| \underset{n \in \mathcal{V}, (X, y_n) \sim \mathcal{Z}}{\mathbb{E}} \left\| \text{out}_n \left( X, A^{1t}, A^{2t} \right) \right\|_2 - \underset{n \in \mathcal{V}, (X, y_n) \sim \mathcal{D}}{\mathbb{E}} \left\| \text{out}_n \left( X, A^{1t}, A^{2t} \right) \right\|_2 \right| \ge \epsilon \right)$$
$$\le \exp\left( \Delta(G)^2 (s\tau_w \|A\|_1)^2 \Omega/8 - s\epsilon\Omega \right) \tag{93}$$

Let $s = \frac{4\epsilon}{\Delta(G)^2 (\tau_w \|A\|_1)^2}$ and $\epsilon = (\tau_w \|A\|_1)^2 \sqrt{\frac{\Delta(G)^4 \log N}{\Omega}}$

$$\mathbb{P}\left( \left| \underset{n \in \mathcal{V}, (X, y_n) \sim \mathcal{Z}}{\mathbb{E}} \left\| \text{out}_n \left( X, A^{1t}, A^{2t} \right) \right\|_2 - \underset{n \in \mathcal{V}, (X, y_n) \sim \mathcal{D}}{\mathbb{E}} \left\| \text{out}_n \left( X, A^{1t}, A^{2t} \right) \right\|_2 \right| \ge \epsilon \right)$$
$$\le \exp\left( -(\tau_w \|A\|_1)^2 \log N \right) \tag{94}$$
$$\le N^{-\tau_w \|A\|_1}$$

with

$$\Omega \ge O\left( \frac{\Delta(G)^2 (\tau_w \|A\|_1)^2 \log N}{\epsilon^2} \right) \tag{95}$$

## D  ADDITIONAL EXPERIMENT

### D.1  ADDITIONAL EXPERIMENT ON SYNTHETIC DATA

**Model and sample complexities with** $\|A^*\|_1$: In Figures 3 (a) (b), we generate a graph $\mathcal{G}$ with $N = 2000$ nodes. $\mathcal{G}$ has one-degree groups and the degree of each node follows a Gaussian distribution $\mathcal{N}(d, \sigma^2)$. $d = 200$. The degrees are truncated to fall within the range of 0 to 500. We vary $A^*$ by changing $\sigma$ and the corresponding $A$. We directly train with $A^*$ to study the impact of $A^*$ on model and sample complexities.

In Figures 3 (a), $|\Omega| = 1200$ and the number of neurons per layer $m$ varies. Fig. 3 (a) shows the testing error decreases as $m$ increases. When $m$ is the same, the testing error increases as $\|A^*\|_1$ increases. This verifies our model complexity in Theorem 3.2. In Figures 3 (b), $m = 50$ and $|\Omega|$ varies. Fig. 3 (b) shows the testing error decreases as $\Omega$ increases. When $\Omega$ is the same, the testing error increases as $\|A^*\|_1$ increases. This verifies our model complexity in Theorem 3.2.

Fig. 3 (c) is a 2D vertical projection of the 3D Fig. 1 (c).

### D.2  ADDITIONAL EXPERIMENT ON REAL DATA

A summary of Ogbn datasets' statistics is presented in Table 2.

Table 2: Ogbn datasets statistics.

| Dataset | Nodes | Edges | Features | Classes | Metric |
|---|---|---|---|---|---|
| Ogbn-Arxiv | 169,343 | 1,166,243 | 128 | 40 | Accuracy |
| Ogbn-Products | 2,449,029 | 61,859,140 | 100 | 47 | Accuracy |

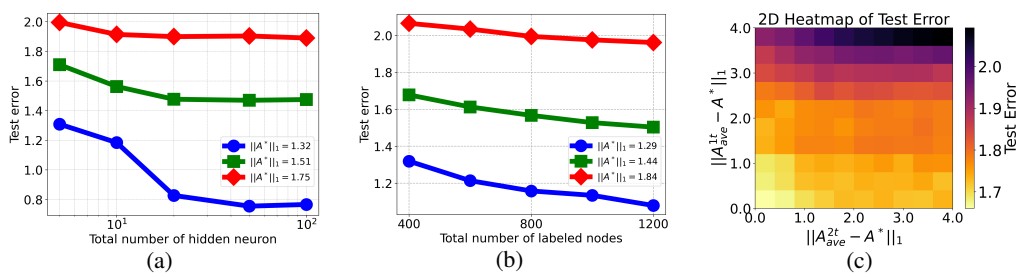

Figure 3: Experiment on synthetic data. (a) Test error with model complexity and $\|A^*\|_1$. (b) Test error with sample complexity and $\|A^*\|_1$. (c) 2D heatmap of 3D Fig. 1 (c)

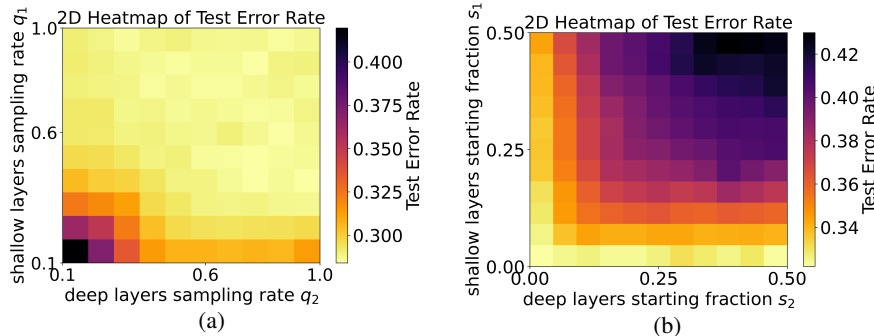

Figure 4: Learning deep GCNs on Ogbn-Arxiv (a) 2D Heatmap of 3D Fig. 2 (a). (b) 2D Heatmap of 3D Fig. 2 (b).

For the Ogbn-Products dataset, we deploy a 4-layer Jumping Knowledge Network (Xu et al., 2018b) GCN with concatenation layer aggregation. Each hidden layer consists of 128 neurons. We define the first two layers as shallow layers and the last two layers as deep layers. with sampling rate $p_2$. The task of Ogbn-Products is to classify the category of a product in a multi-class, where the 47 top-level categories are used for target labels. We use 60% of the data for training, 20% for testing, and 20% for verification.

We run the similar experiment as Figure 2 (a) for Obgn-Arxiv dataset. We fix $q_1 = 0.1$ and vary $q_2$ from 0.1 to 1.0 at increments of 0.1, then fix $q_2 = 0.1$ and vary $q_1$ from 0.1 to 1.0 at increments of 0.1. Figure 5 shows that with the increasing sampling rate in shallow layers, the test accuracy is higher than the test accuracy with the increasing sampling rate in deep layers. It suggests that the generalization is more sensitive to the sampling in the shallow layers rather than deep layers, consistent with observations in other datasets.

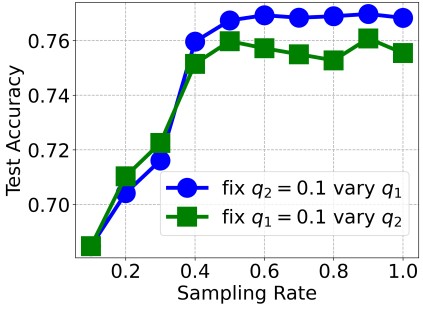

Figure 5: Layer-wise Sampling Rate Effect on Ogbn-Products

