# OpenReview forum: "How do skip connections affect Graph Convolutional  networks  with graph sampling? A theoretical analysis on generalization"
_ICLR.cc/2024/Conference — Submitted to ICLR 2024_

### Official Review · Reviewer_YYVW · 2023-10-13

**Soundness:** 1 poor
**Presentation:** 1 poor
**Contribution:** 1 poor
**Rating:** 5
**Confidence:** 5

**Summary:**

This paper delves into the generalization of a two-layer GNN that takes both the residual connection and sampling into consideration. This paper is very hard to digest due to writting issue.

**Strengths:**

The orginal motivation of this paper is very interesting and important. But the theoritical part needs improvment.

**Weaknesses:**

The paper looks fine until Section 3.3.

However, beyond Section 3.3, it gives the impression that the authors may not have fully understand Allen-Zhu & Li's work before extending it from a residual-connected MLP to a residual-connected GCN. Consequently, the notations, proofs, and overall presentation become challenging to digest:

1. The presentation lacks clarity, with numerous notations introduced without proper explanation. For instance, in the paragraph surrounding Equations 6 and 7, the authors transform the original 2-layer residual-connected GCN into a new format, but it remains unclear how these two formulations correspond. Questions arise regarding the meaning of symbols like $p\_\mathcal{F}$ and $p\_\mathcal{G}$. Additionally, it's unclear what $\sum_{i\in [p\_\mathcal{F}]}$ and $\sum_{i\in [p\_\mathcal{G}]}$ represent in the context of GCN. It's also unclear whether $w_{r,i}^\star$ refers to the i-th or r-th row/column of a weight matrix. This section is not well organized, making it exceptionally challenging to comprehend.

2. The notation on Page 6, particularly at the top under the algorithm section, is even more convoluted. The authors introduce numerous notations, but their purpose and connection to GCN are unclear. I have no idea how these notations relate to the GCN.

3. There appear to be errors in the paper. Detailed questions and concerns have been raised, which should be addressed for clarity and accuracy.

4. 2-layer GNN is not deep enough comparing to existing works. Especially the authors argue existing works' considered model is not deep ...

**Questions:**

1. On Page 5, at the top, we have this $d_1 \sqrt{d_i/d_j}$. Where does this $d_1$ originate from? Why does the node sampling probability always depend on this value?

2. According to your sampling method, neither $A^\star$ nor the "expectation of the sampled adjacency matrices" is identical to the original adjacency matrix $A$. Won't this introduce bias during training? In other words, your objective differs from the original objective function. In this case, how can we ensure generalization?

3. What does the first term on page 6, in the first line, represent? Without a clear explanation, I cannot grasp the impact of $A^\star$ on the $\epsilon_0$ of Thm3.2.

4. Why C is not trainable in Eq. 1-4? Since this is linear-regression using square loss, cannot we just think this C as identity matrix?

5. What is the $r_w$ and $r_v$ in Eq. 11-12?

---

> ### Author Response · Authors · 2023-11-23
>
> ## Weakness 1:
>  The presentation lacks clarity, with numerous notations introduced without proper explanation. For instance, in the paragraph surrounding Equations 6 and 7, the authors transform the original 2-layer residual-connected GCN into a new format, but it remains unclear how these two formulations correspond. Questions arise regarding the meaning of symbols like $p_\mathcal{F}$ and $p_\mathcal{G}$. Additionally, it's unclear what $\sum_{i \in [p_\mathcal{F}]}$ and $\sum_{i \in [p_\mathcal{G}]}$ represent in the context of GCN. It's also unclear whether $w_{r,i}^{*}$ refers to the i-th or r-th row/column of a weight matrix. This section is not well organized, making it exceptionally challenging to comprehend.
>
> ## Answer Weakness 1:
> **The role of equations (6) and (7)**. We first need to clarify that we do not transform the learner network, which is 2-layer residual-connected GCN, to equations (6) and (7). The concept class in (6) and (7) is introduced for the theoretical generalization analysis only. We first define many target functions in the form of (6) and (7) to predict the labels. These functions have the same form but different parameters of $a_{\mathcal{F}, r, i}^*$, $a_{\mathcal{G}, r, i}^*$, $w^*_{r,i}$, $v_{r,i}^*$. Then OPT is defined as the smallest label prediction error of the best target function  (over the choices of parameters) in the concept class. OPT decreases as the concept class becomes more complex, such as increasing the integers $p_\mathcal{F}, p_\mathcal{G}$ and making activations  $\mathcal{F}$ and $\mathcal{G}$ more complex.  Note that the definition of OPT itself is irrelevant to the GCN learning, it is purely defined from an approximation perspective.
>
>   Then when we analyze learning 2-layer GCN, we show that the label prediction error by the learned 2-layer GCN model is close to $10$ OPT. Intuitively, that means the learned GCN model can  approximate almost the best target function in the form of (6) and (7).
>
>   This approach, i.e., first defining target functions to approximate labels and then quantifying generalization performance with respect to  the best target function,  follows from those in xxxxxx. (cite Allen Zhu's and Hongkang's paper).
>
> **The meaning of  $p_\mathcal{F}, p_\mathcal{G}$, $\sum_{i \in [p_\mathcal{F}]}$**
> $p_\mathcal{F}, p_\mathcal{G}$ are two positive integers. We want to say $F_{A^*}$ and $G_{A^*}$ are sum of  $p_\mathcal{F}$ and $p_\mathcal{G}$ functions respectively.  We realized that the notation of $\sum_{i \in [p_\mathcal{F}]}$ $\sum_{i \in [p_\mathcal{G}]}$can be confusing.
> We revised them to $\sum_{i=1}^{p_\mathcal{F}}$ and $\sum_{i=1}^{p_\mathcal{G}}$.
>
> In $a_{\mathcal{F}, r, i}^*, a_{\mathcal{G}, r, i}^*$, $w_{r, i}^*$, $v_{r, i}^*$, the subscript $r, i$ are not indices for vectors or matrices. We use $r,i$ to indicate that for every function $i$ and the $r$th row, there are different sets of coefficients $a_{\mathcal{F}}$, $a_{\mathcal{G}}$, vectors $w^*$ and $v^*$. For example, for ever $r$ and $i$, $w_{r, i}^*$ is a vector in  $\mathbb{R}^{d}$,  and  $v_{r, i}^*$ is a vector in $\mathbb{R}^{k}$. We revised in the paper that accordingly that
>
>   ``Given $r$, $i$, the scalers $a_{\mathcal{F}, r, i}^*, a_{\mathcal{G}, r, i}^*$ are in $[-1,1]$, the vectors     $w_{r, i}^*$ are in  $\mathbb{R}^{d}$,  and  $v_{r, i}^*$ are in $\mathbb{R}^{k} $. For simplicity, we assume $\left \| w_{r, i}^* \right \|=\left \| v_{r, i}^* \right \| =\frac{1}{\sqrt{2} } $ for all $r$, $i$.  ''
>  We have revised this section, with major changes highlighted.
>
> ##  Weakness 2:
> The notation on Page 6, particularly at the top under the algorithm section, is even more convoluted. The authors introduce numerous notations, but their purpose and connection to GCN are unclear. I have no idea how these notations relate to the GCN.
>
> ##  Answer Weakness 2:
> We realized that some notations are not necessary or can be simplified for presenting the main theorem. We removed some unnecessary notations and added the formal definition of model complexity constants $C_{\varepsilon}$ and sample complexity constants $C_{\mathfrak{s}}$ in (8) and (9). We also simplified the assumptions of Theorem 3.2 with necessary notations only.
>
> ## Weakness 3:
> There appear to be errors in the paper. Detailed questions and concerns have been raised, which should be addressed for clarity and accuracy.
>
> ## Answer Weakness 3:
> We have addressed the specific comments in the following questions.

---

> ### Author Response · Authors · 2023-11-23
>
> ## Weakness 4:
> 2-layer GNN is not deep enough comparing to existing works. Especially the authors argue existing works' considered model is not deep ...
>
> ## Answer Weakness 4:
> We understand that the reviewer would like to see a theoretical analysis of a network model that is more complicated than the current one. We agree that this is a valid point, and we would like to explore that direction in the future.  However, we first need to stress that **the simple model we consider in this paper is already advancing the state-of-the-art model for the generalization analysis for graph neural networks**. For example, recent works [2, 6] are published in ICML and NIPS that build the theoretical analysis on two-layer neural networks. In fact, the work [1] only considers one-layer GCNs.  Only [5] consider deeper networks, but they typically do not include non-linear activation functions.
>
> 1. To the best of our knowledge, the only approach that considers graph neural networks, or even the general neural networks with more than two layers theoretically, is the neural tangent kernel (NTK) approach. However, the  NTK approach considers the scenario that the neural network stays close to the linearization around the initialization and does not characterize the nonlinear activations.
> Therefore, there is a performance gap between practical neural networks and the NTK approach, as shown in [3, 4]. Thus, we do not follow this line. In contrast, we directly analyze the interactions of nonlinear activations across layers.
>
> 2. We need to emphasize that **our paper makes novel theoretical contributions**, despite the two-layer model. Specifically,   our paper provides the first theoretical analysis of GCNs with skip connection using graph sampling.  We discover that skip connections necessitate varying sampling requirements across different layers. This finding challenges the conventional sampling approach that is uniform to different layers and suggests the need for layer-specific sampling considerations.
>
> 3. **We have verified our theoretical insights in deep GCNs on large datasets.** For example, in Figure 2(a), we train an 8-layer Jumping Knowledge Network   GCN with concatenation layer aggregation on the  Ogbn-Arxiv dataset.  We verify in this deep GCN that graph sampling in shallow layers has a more significant impact than graph sampling in deeper layers.
>
> ## Reference
> 1. Zhang, Shuai, et al. "How unlabeled data improve generalization in self-training? A one-hidden-layer theoretical analysis." International Conference on Learning Representations. 2021.
> 2. Hongkang Li, Meng Wang, Sijia Liu, Pin-Yu Chen, and Jinjun Xiong. Generalization guarantee of
> training graph convolutional networks with graph topology sampling. In International Conference
> on Machine Learning (ICML), pp. 13014–13051. PMLR, 2022a.
> 3. Chen, Shuxiao, Hangfeng He, and Weijie Su. "Label-aware neural tangent kernel: Toward better generalization and local elasticity." Advances in Neural Information Processing Systems 33 (2020): 15847-15858.
> 4. Arora, Sanjeev, et al. "Harnessing the power of infinitely wide deep nets on small-data tasks." arXiv preprint arXiv:1910.01663 (2019).
> 5. Keyulu Xu, Mengshi Zhang, Stefanie Jegelka, and Kenji Kawaguchi. Optimization of graph neural
> networks: Implicit acceleration by skip connections and more depth. In International Conference
> on Machine Learning (ICML). PMLR, 2021.
> 6. Cao, Yuan, et al. "Benign overfitting in two-layer convolutional neural networks." Advances in neural information processing systems 35 (2022): 25237-25250.

---

> ### Author Response · Authors · 2023-11-23
>
> ## Question 1:
> On Page 5, at the top, we have this $d_1 \sqrt{\frac{d_i}{d_j}}$. Where does this $d_1$ originate from? Why does the node sampling probability always depend on this value?
>
> ## Answer Question 1:
> 1. We follow the same assumption on node degrees as that in [1]. Specifically, the node
> degrees can be divided into L   groups, with each group having $N_l$ nodes.   The degrees of all these $N_l$ nodes in group $l$ are in the order of $d_l$, i.e., between $cd_l$ and $Cd_l$ for
> some constants $c \leq  C$. $d_l$ is order-wise smaller than $d_{l+1}$, i.e., $d_l = o(d_{l+1})$. Then $d_1$ is the smallest degree (order-wise) of these degree groups.
>
> 2. Our core sampling concept revolves around sampling edges between nodes of lower degrees with higher probability. Because the edges between lower-degree nodes correspond to larger entries in $A$, we want to sample larger entries with a higher probability and smaller entries with a lower probability. In our sampling strategy, we just divide entries in $A$ into two parts, the parts with larger values are sampled with $1-p_{ij}^k$, and the group with smaller values is sampled with $p_{ij}^k$, where $p_{ij}^k$. The size of the larger part, which is selected as $d_1\sqrt{d_i/d_j}$, is selected to ensure that $ \left \| A^* \right \| _1 $ is $O(1)$, which is required in our generalization analysis. It does not have to the exact value of $d_1\sqrt{d_i/d_j}$, any value in the order of $d_1\sqrt{d_i/d_j}$ will not change our order-wise analysis.
>
> 3. We want to emphasize that although we set this value to simplify our theoretical analysis, **the main idea of
> sampling edges between nodes of lower degrees with higher probability are preserved in our sampling strategy.**
>
> ## Question 2:
> According to your sampling method, neither $A^*$ nor the "expectation of the sampled adjacency matrices" is identical to the original adjacency matrix $A$. Won't this introduce bias during training? In other words, your objective differs from the original objective function. In this case, how can we ensure generalization?
>
> ## Answer Question 2:
> 1. We agree with the statement that neither $A^*$ nor the "expectation of the sampled adjacency matrices" is identical to the original adjacency matrix $A$.'' That is indeed one of our main messages that graph sampling does not have to get an unbiased estimator of $A$. We only need to make sure the resulting $A^*$ can accurately characterize the data correlation. $A^*$ can be sparse than $A$. Because the original adjacency matrix $A$ of the graph often contains redundant information, learning using   $ A^* $ can perform comparable to, or even better than $ A $. That explains why graph sampling can achieve desirable generalization.
>
> 2. We also verified the redundancy and the effectiveness of $A^*$ empirically.  In  Figure 2a,  when both $ q_1 $ and $ q_2 $ are above 0.6, which corresponds to sparse effective adjacency matrix $ A^* $, the test error aligns closely with results using the original $ A $ (where $ q_1 $ and $ q_2 $ are 1). This implies that our model, utilizing roughly 60\% of the graph, performs comparably to using the full graph.
>
> ## Question 3:
> What does the first term on page 6, in the first line, represent? Without a clear explanation, I cannot grasp the impact of $A^*$ on the $\epsilon_0$ of Thm3.2.
>
> ## Answer Question 3:
> 1. We included the definition of the model complexity constants $C_{\varepsilon}$ and sample complexity constants $C_\mathfrak{s}$ in (8) and (9). The model complexity constants $\mathfrak{C}_\varepsilon$ and sample complexity constants $\mathfrak{C}_s$ those in    \cite{li2022generalization} (Section 1.2) and  \cite{NEURIPS2019_5857d68c} (Section 4). They represent the required number of model parameters and training samples to learn $\phi$ up to $\epsilon$ error, respectively.
>
> 2.  When $\left \| A^* \right \|_1 $ increases, both the model complexity constants and sample complexity constants of functions $\mathcal{F}$ and $\mathcal{G}$ incease. Then $\epsilon_0$ increases, indicating a larger generalization error. Theorem 3.2 also indicates the model complexity $M_0$ (the number of neurons)  and the sample complexity $N_0$ (the number of labeled nodes) both increase. We added this discussion to the paper.
>
> ## Reference
> 1. Li, Hongkang, et al. "Generalization guarantee of training graph convolutional networks with graph topology sampling." International Conference on Machine Learning. PMLR, 2022.

---

> ### Author Response · Authors · 2023-11-23
>
> ## Question 4:
> Why $C$ is not trainable in Eq. 1-4? Since this is linear-regression using square loss, cannot we just think this $C$ as identity matrix?
>
> ## Answer Question 4:
> 1. We would like to clarify that an untrained output is a typical setting for theoretical analysis for generalization.  Existing works [1, 2, 3] on top conferences also made this assumption in their analyses.
>
> 2. In our study, the decision to sample $C_{i,j}$ from $\mathcal{N}(0, 1/m)$ is primarily to simplify the analysis, ensuring that the norm $\left \| C \right \| \leq 1$, as detailed in Appendix section B.1.
> Furthermore, it is indeed feasible to consider $C$ as an identity matrix scaled by $1/m$. This alternative representation would effectively function as a mean pooling layer, providing a similar effect while maintaining the simplicity and boundedness required for theoretical analysis.
>
> ## Question 5:
> What is the $r_w$ and $r_v$ in Eq. 11-12?
>
> ## Answer Question 5:
> I think you mean  $\tau_w$ and $ \tau_v $. We prove that $ \|W_t\|_2 $ and $ \|V_t\|_2 $  are bounded by $\tau_w$ and $\tau_v$ in Appendix section B.3. Because $\tau_w$ and $\tau_v$ can be bounded by model complexity and sample complexity constants to simply the presentation of the paper, we removed $\tau_w$ and $\tau_v$ from the main text and replace them with their bounds using $ C_F  $ and $ C_G  $.
>
> ## Reference
> 1. Allen-Zhu, Zeyuan, and Yuanzhi Li. "What can resnet learn efficiently, going beyond kernels?." Advances in Neural Information Processing Systems 32 (2019).
> 2. Li, Hongkang, et al. "Generalization guarantee of training graph convolutional networks with graph topology sampling." International Conference on Machine Learning. PMLR, 2022.
> 3. Zhang, Shuai, et al. "Joint Edge-Model Sparse Learning is Provably Efficient for Graph Neural Networks." In International Conference on Learning (ICLR), 2023.

---

### Official Review · Reviewer_cdgo · 2023-11-01

**Soundness:** 4 excellent
**Presentation:** 3 good
**Contribution:** 2 fair
**Rating:** 5
**Confidence:** 4

**Summary:**

In this paper, the authors try to figure the relationship between graph sampling and skip-connections in GCN. Based on this motivation, many solid theories have been given. Also they validate the theoretical results on benchmark datasets.

**Strengths:**

1. This paper gives a different theoretical perspective on the relationship between graph sampling and different layers of GCNs.

2. It provides very solid theoretical analysis. It is convincing.

**Weaknesses:**

This work is far from the real setting of graph neural networks.
1. In this paper, they assume all with perfectly homophilous graphs. Actually, it is not going to happen in heterophilous graphs. If we have perfectly homophilous graphs, the stationary point will make the generalization happen.

2. The setting of two-layer skip connections is oversimple. Two graph convolutions only access two-hop neighborhoods. Thus, I cannot imagine how these conclusions can inspire this community (Graph Neural Network).

3. The dense graph is not a usual condition. Even the transformer generates an implicit graph from batch data (then it is sparse from the global point.). If the graph is dense and perfectly homophilous, then the graph convolution basically equals making every node its' class center.

**Questions:**

I have some concerns about how this work can help graph neural network (or transformer) community.

---

> ### Author Response · Authors · 2023-11-23
>
> ## Weakness 1:
> In this paper, they assume all with perfectly homophilous graphs. Actually, it is not going to happen in heterophilous graphs. If we have perfectly homophilous graphs, the stationary point will make the generalization happen.
> ## Answer Weakness 1:
> 1. We need to emphasize that **reaching a stationary point of the training optimization problem does not automatically lead to generalization**. Overfitting is a typical counterexample. If a model is overly complex or trained with insufficient sample complexity, it might overfit the training data to achieve zero training loss but fail to generalize well to new data.
> 2. **We do not assume homophilous graphs in this paper, and our theoretical results apply to both homophilous graphs and heterophilous graphs**. Our main result Theorem 3.2 indicates that the learned model can predict labels with a risk about 10 OPT, where OPT is the smallest approximation error of the target function in the form of (6) and (7). It is possible that, when everything else is the same,  OPT is smaller if the graph is homophilous rather than heterophilous, but our Theorem 3.2 holds despite whether OPT is small or large. Moreover, our insight in Lemma 3.1 that graph sampling in different layers contributes to the output approximation differently. This sight applies to both homophilous graphs and heterophilous graphs.
> 3. We agree with the reviewer that GCNs have been mostly employed in homophilous graphs in the literature, maybe because they have better generalization performance in homophilous graphs. However, our results do not require the assumptions of homophilous graphs and can characterize the GCN generalization performance (no matter it is good or bad) in both homophilous graphs and heterophilous graphs, as discussed in the previous point.
>
> ## Weakness 3:
> The dense graph is not a usual condition. Even the transformer generates an implicit graph from batch data (then it is sparse from the global point.). If the graph is dense and perfectly homophilous, then the graph convolution basically equals making every node its' class center.
> ## Answer Weakness 3:
> We do not assume the graph is dense. We realize this confusion may result from our statement in the original submission that
> "This research illustrates that the target functions rely on $A^*$ instead of the original graph's adjacency matrix $A$." What we should have said is that $A^*$ can be sparser than $A$. We did not assume that the graph represented by $A$ is inherently dense or sparse. Consequently, we have revised the original sentence to state that “$A^*$ can be sparser than $A$.”

---

> ### Author Response · Authors · 2023-11-23
>
> ## Weakness 2:
> The setting of two-layer skip connections is oversimple. Two graph convolutions only access two-hop neighborhoods. Thus, I cannot imagine how these conclusions can inspire this community (Graph Neural Network).
> ## Answer Weakness 2:
> We understand that the reviewer would like to see a theoretical analysis of a network model that is more complicated than the current one. We agree that this is a valid point, and we would like to explore that direction in the future.  However, we first need to stress that **the simple model we consider in this paper is already advancing the state-of-the-art model for the generalization analysis for graph neural networks**. For example, recent works [2, 6] are published in ICML and NIPS that build the theoretical analysis on two-layer neural networks. In fact, the work [1] only considers one-layer GCNs.  Only [5] consider deeper networks, but they typically do not include non-linear activation functions.
>
> 1. To the best of our knowledge, the only approach that considers graph neural networks, or even the general neural networks with more than two layers theoretically, is the neural tangent kernel (NTK) approach. However, the  NTK approach considers the scenario that the neural network stays close to the linearization around the initialization and does not characterize the nonlinear activations.
> Therefore, there is a performance gap between practical neural networks and the NTK approach, as shown in [3, 4]. Thus, we do not follow this line. In contrast, we directly analyze the interactions of nonlinear activations across layers.
>
> 2. We need to emphasize that **our paper makes novel theoretical contributions**, despite the two-layer model. Specifically,   our paper provides the first theoretical analysis of GCNs with skip connection using graph sampling.  We discover that skip connections necessitate varying sampling requirements across different layers. This finding challenges the conventional sampling approach that is uniform to different layers and suggests the need for layer-specific sampling considerations.
>
> 3. **We have verified our theoretical insights in deep GCNs on large datasets.** For example, in Figure 2(a), we train an 8-layer Jumping Knowledge Network   GCN with concatenation layer aggregation on the  Ogbn-Arxiv dataset.  We verify in this deep GCN that graph sampling in shallow layers has a more significant impact than graph sampling in deeper layers.
>
> ## Reference
> 1. Zhang, Shuai, et al. "How unlabeled data improve generalization in self-training? A one-hidden-layer theoretical analysis." International Conference on Learning Representations. 2021.
> 2. Hongkang Li, Meng Wang, Sijia Liu, Pin-Yu Chen, and Jinjun Xiong. Generalization guarantee of
> training graph convolutional networks with graph topology sampling. In International Conference
> on Machine Learning (ICML), pp. 13014–13051. PMLR, 2022a.
> 3. Chen, Shuxiao, Hangfeng He, and Weijie Su. "Label-aware neural tangent kernel: Toward better generalization and local elasticity." Advances in Neural Information Processing Systems 33 (2020): 15847-15858.
> 4. Arora, Sanjeev, et al. "Harnessing the power of infinitely wide deep nets on small-data tasks." arXiv preprint arXiv:1910.01663 (2019).
> 5. Keyulu Xu, Mengshi Zhang, Stefanie Jegelka, and Kenji Kawaguchi. Optimization of graph neural
> networks: Implicit acceleration by skip connections and more depth. In International Conference
> on Machine Learning (ICML). PMLR, 2021.
> 6. Cao, Yuan, et al. "Benign overfitting in two-layer convolutional neural networks." Advances in neural information processing systems 35 (2022): 25237-25250.
>
> ## Answer Question 1:
> I think we can answer this question in Weakness 2.

---

### Official Review · Reviewer_8ATu · 2023-11-07

**Soundness:** 2 fair
**Presentation:** 1 poor
**Contribution:** 2 fair
**Rating:** 3
**Confidence:** 2

**Summary:**

The authors analyze the effect of subsampling the symmetrically normalized adjacency matrix in node regression tasks for two layered GCNs with a single skip connection. Arguing that the second layer is less affected by a "bad" sampling than the first layer, the authors propose an algorithm that samples different matrices for different layers. The sampling strategy is based on sampling edges with higher probability from low-degree nodes. Finally, the authors provide theoretical results for GCNs trained with SGD that show sample complexity bounds to achieve near-optimal performance.

**Strengths:**

- The analyzed problem itself is highly interesting; While many papers focus on expressivity, this papers gives generalization and complexity bounds for GNNs, which is a hard and interesting problem.
- The experimental results seem to match the theory well
- Proofs backup theoretical results

**Weaknesses:**

- The authors tend to overstate their results, for example, it is often mentioned that the work would analyze a large class of graph learning models, while only two-layer GCNs are analyzed, where the final prediction layer is not trained. This is not a realistic scenario. While frequently, the GCN layers are not trained the final layer is, up to the Reviewers' knowledge, always trained to be able to linearly separate classes.
- Another example is in Section 3.1, where the authors highlight their own work in comparison to others by mentioning that other works only accommodate "shallow GCNs". However, the cited works also analyze two-layered GCNs.
- Some of the assumptions seem highly restrictive: For example in section 3.3. the function $\mathcal{F}$ and $\mathcal{G}$ are assumed to be smooth functions on $\mathbb{R}^{d \times N} \times \mathbb{R}^{N \times N}$. However, as the domain is the graph domain, it is not clear whether this is a reasonable assumption as this could break the permutation equivariance.
- Another example is the final (untrained) layer in Equation 5, which is simply a matrix multiplication. Thus, does not satisfy universal approximation properties.
- Many notations and definitions are missing, which leads to confusion. For example, Section 3.4 is unclear.
- The work lacks clarity, and often explanations and intuitions are not given. For example, in their main results, Lemma 3.1 and Theorem 3.2 the assumptions of the results are not clear. The results are also not well-presented.

**Questions:**

While the work analyzes an interesting theoretical question, the writing and presentation lack clarity and mathematical preciseness. Which are necessary to be able to value the presented results. I would recommend the authors to go over their work again, and make sure that every notion is well-defined and intuitions are given.

Some more Questions:
-  What is OPT in Equation 13? $\mathcal{H}_{n,A^*}$ is defined as the target function, while $y_n$ is the label of node $n$. How is it possible that $OPT$ is non-zero?
-  In Lemma 3.1 and Theorem 3.2: It is not clear with respect to which event the probability is taken.
- In Theorem 3.2: Could the authors present the assumptions better or give more intuitions?
- Why do the authors average in Equation 14 over all iterations of the SGD steps?
- How is $(X,y_n)$ sampled, and how is $\mathcal{D}$ defined?
- Could you elaborate on many assumptions, e.g., Section 3.4: it doesn't seem clear that the norms of the learned weight matrices are uniformly bounded.

---

> ### Author Response · Authors · 2023-11-23
>
> ## Weakness 1:
> The authors tend to overstate their results, for example, it is often mentioned that the work would analyze a large class of graph learning models, while only two-layer GCNs are analyzed, where the final prediction layer is not trained. This is not a realistic scenario. While frequently, the GCN layers are not trained the final layer is, up to the Reviewers' knowledge, always trained to be able to linearly separate classes.
> ## Answer Weakness 1:
> We would like to clarify that an untrained output is a typical setting for theoretical analysis for generalization. Existing works [1, 2, 3, 4, 5] on top conferences also made this assumption in their analyses. **Meanwhile, although we make such an assumption, the problem we solve is still challenging and significant.**
> 1. Existing works on the generalization of GNNs cannot characterize the influence of graph sampling with skip connections.
> 2. Our paper provides the first theoretical analysis of GCNs with skip connection using graph sampling. We discover that skip connections necessitate varying sampling requirements across different layers. This finding challenges the conventional sampling approach that is uniform to different layers and suggests the need for layer-specific sampling considerations.
>
> ## Reference
> 1. Allen-Zhu, Zeyuan, Yuanzhi Li, and Yingyu Liang. "Learning and generalization in overparameterized neural networks, going beyond two layers." Advances in neural information processing systems 32 (2019).
> 2. Allen-Zhu, Zeyuan, Yuanzhi Li, and Zhao Song. "A convergence theory for deep learning via over-parameterization." International conference on machine learning. PMLR, 2019.
> 3. Cao, Yuan, et al. "Benign overfitting in two-layer convolutional neural networks." Advances in neural information processing systems 35 (2022): 25237-25250.
> 4. Li, Hongkang, et al. "Generalization guarantee of training graph convolutional networks with graph topology sampling." International Conference on Machine Learning. PMLR, 2022.
> 5. Zhang, Shuai, et al. "Joint Edge-Model Sparse Learning is Provably Efficient for Graph Neural Networks." In International Conference on Learning (ICLR), 2023.
>
>
> ## Weakness 3:
> Some of the assumptions seem highly restrictive: For example in section 3.3, the function $\mathcal{F}$ and $\mathcal{G}$ are assumed to be smooth functions on $\mathbb{R}^{d \times N} \times \mathbb{R}^{N \times N}$. However, as the domain is the graph domain, it is not clear whether this is a reasonable assumption as this could break the permutation equivariance.
>
>
> ## Answer Weakness 3:
> We want to clarify that our concept class of target functions satisfies the permutation equivariance property. That is because when we permutate the node indices, the corresponding adolescences matrix is also permuted. To see this, consider a toy example of a two-node graph. }
> Let $N=2$, given the feature matrix ($X\in\mathbb{R}^{d \times 2}$ ) and adjacency matrix ( $A\in\mathbb{R}^{2 \times 2}$ ):
> $$
> X = \left[ \begin{array}{cc}
> x_1 & x_2
> \end{array} \right],
> A = \left[ \begin{array}{cc}
> a_{11} & a_{12} \\
> a_{21} & a_{22}
> \end{array} \right]
> $$
>
>
>
> Applying the function \( f \) to the product \( XA \) gives:
> $$
>  f(XA) = \left[ \begin{array}{cc}
> f(a_{11}x_1 + a_{12}x_2), & f(a_{21}x_1 + a_{22}x_2)
> \end{array} \right]
> $$
> where the first entry of $f(XA)$ corresponds to the output of $x_1$, and the first entry of $f(XA)$ corresponds to the output of $x_2$.
>
> If we swap the node indices of these two nodes such that $x_1=x_2$ and $x_2=x_1$, then the feature matrix and the new adjacency matrix become
> $$
>  X' = \left[ \begin{array}{cc}
> x_2, & x_1
> \end{array} \right],
> A' = \left[ \begin{array}{cc}
> a_{22}, & a_{21} \\
> a_{12}, & a_{11}
> \end{array} \right]
> $$
> The function $ f $  applied to the product $ X'A' $  results in:
> $$
> f(X'A') = \left[ \begin{array}{cc}
> f(a_{12}x_1 + a_{22}x_2), & f(a_{11}x_1 + a_{21}x_2)
> \end{array} \right]
> $$
> where the first entry of $f(X'A')$ corresponds to the output of $x'_1$, which equals $x_2$, and the first entry of $f(X'A')$ corresponds to the output of $x'_2$, which is $x_1$.
> This demonstrates that $ f $ is permutation equivariant.

---

> ### Author Response · Authors · 2023-11-23
>
> ## Weakness 4:
> Another example is the final (untrained) layer in Equation 5, which is simply a matrix multiplication. Thus, does not satisfy universal approximation properties.
> ## Answer Weakness 4:
> As we said in Weaknesses 1, we would like to clarify that an untrained output is a typical setting for theoretical analysis for generalization. Existing works on top conferences also made this assumption in their analyses.
>
> The universal approximation property [1, 2] refers to the expressive power of a multi-layer feedforward neural network that the network can approximate any real-valued continuous function to any desired degree of accuracy. We agree that the network in Equation 5 may not be able to approximate all the possible continuous functions. **However, our work focuses on analyzing the convergence and generalization of GNN with skip connections, which is a different aspect from function approximation.** Existing works that study the neural network using the universal approximation property usually do not involve the analysis of convergence and generalization. Our analysis shows that learning a two-hidden-layer GNN with a skip connection using graph sampling can achieve a label prediction error similar to the best prediction error among a concept class of target functions.
>
> ## Reference
> 1. Hornik, Kurt, Maxwell Stinchcombe, and Halbert White. "Multilayer feedforward networks are universal approximators." Neural networks 2.5 (1989): 359-366.
> 2. Lu, Zhou, et al. "The expressive power of neural networks: A view from the width." Advances in neural information processing systems 30 (2017).
>
> ## Weakness 5:
> Many notations and definitions are missing, which leads to confusion. For example, Section 3.4 is unclear.
> ## Answer Weakness 5:
> We have already simplified our notations and definitions and deleted some complex functions.
>
> ## Weakness 6:
> The work lacks clarity, and often explanations and intuitions are not given. For example, in their main results, Lemma 3.1 and Theorem 3.2 the assumptions of the results are not clear. The results are also not well-presented.
>
> ## Answer Weakness 6:
> We have simplified the assumptions in Lemma 3.1 and Theorem 3.2 to improve the presentation.  We update  Lemma 3.1 and Theorem 3.2.
>
> Moreover, to clarify the assumption of Lemma 3.1, we added the following to the paper
> ``Note that $\widetilde{\Theta}\left(\alpha  C_\mathfrak{s}(\mathcal{G})\right)<1$, then the upper bound for $p_{ij}^2$ is higher than that for $p_{ij}^1$ in the assumption. That means the sampling for the first hidden layer must focus more on low-degree edges, while such a requirement is  relaxed in the second layer.''
> To better interpret Theorem 3.2, we added the following discussion,
> "Moreover, when $\|A^*\|_1$ increases,  and  $C_s$, and $\epsilon_0$ all increase.  Theorem 3.2  indicates the model complexity $M_0$,  $N_0$, and the generalization error $\epsilon$ all increasing, indicating worse prediction performance."

---

> ### Author Response · Authors · 2023-11-23
>
> ## Weakness 2:
> Another example is in Section 3.1, where the authors highlight their own work in comparison to others by mentioning that other works only accommodate "shallow GCNs". However, the cited works also analyze two-layered GCNs.
> ## Answer Weakness 2:
> We understand that the reviewer would like to see a theoretical analysis of a network model that is more complicated than the current one. We agree that this is a valid point, and we would like to explore that direction in the future.  However, we first need to stress that **the simple model we consider in this paper is already advancing the state-of-the-art model for the generalization analysis for graph neural networks**. For example, recent works [2, 6] are published in ICML and NIPS that build the theoretical analysis on two-layer neural networks. In fact, the work [1] only considers one-layer GCNs.  Only [5] consider deeper networks, but they typically do not include non-linear activation functions.
>
> 1. To the best of our knowledge, the only approach that considers graph neural networks, or even the general neural networks with more than two layers theoretically, is the neural tangent kernel (NTK) approach. However, the  NTK approach considers the scenario that the neural network stays close to the linearization around the initialization and does not characterize the nonlinear activations.
> Therefore, there is a performance gap between practical neural networks and the NTK approach, as shown in [3, 4]. Thus, we do not follow this line. In contrast, we directly analyze the interactions of nonlinear activations across layers.
>
> 2. We need to emphasize that **our paper makes novel theoretical contributions**, despite the two-layer model. Specifically,   our paper provides the first theoretical analysis of GCNs with skip connection using graph sampling.  We discover that skip connections necessitate varying sampling requirements across different layers. This finding challenges the conventional sampling approach that is uniform to different layers and suggests the need for layer-specific sampling considerations.
>
> 3. **We have verified our theoretical insights in deep GCNs on large datasets.** For example, in Figure 2(a), we train an 8-layer Jumping Knowledge Network   GCN with concatenation layer aggregation on the  Ogbn-Arxiv dataset.  We verify in this deep GCN that graph sampling in shallow layers has a more significant impact than graph sampling in deeper layers.
>
> ## Reference
> 1. Zhang, Shuai, et al. "How unlabeled data improve generalization in self-training? A one-hidden-layer theoretical analysis." International Conference on Learning Representations. 2021.
> 2. Hongkang Li, Meng Wang, Sijia Liu, Pin-Yu Chen, and Jinjun Xiong. Generalization guarantee of
> training graph convolutional networks with graph topology sampling. In International Conference
> on Machine Learning (ICML), pp. 13014–13051. PMLR, 2022a.
> 3. Chen, Shuxiao, Hangfeng He, and Weijie Su. "Label-aware neural tangent kernel: Toward better generalization and local elasticity." Advances in Neural Information Processing Systems 33 (2020): 15847-15858.
> 4. Arora, Sanjeev, et al. "Harnessing the power of infinitely wide deep nets on small-data tasks." arXiv preprint arXiv:1910.01663 (2019).
> 5. Keyulu Xu, Mengshi Zhang, Stefanie Jegelka, and Kenji Kawaguchi. Optimization of graph neural
> networks: Implicit acceleration by skip connections and more depth. In International Conference
> on Machine Learning (ICML). PMLR, 2021.
> 6. Cao, Yuan, et al. "Benign overfitting in two-layer convolutional neural networks." Advances in neural information processing systems 35 (2022): 25237-25250.

---

> ### Author Response · Authors · 2023-11-23
>
> ## Question 1:
> What is OPT in Equation 13? $ \mathcal{H}_{n, A^*} $ is defined as the target function, while $ y_n $ is the label of node $ n $. How is it possible that OPT is non-zero?
>
> ## Answer Question 1:
> 1. We revised the  OPT definition, which is in equation (11) of the revision  to resolve the confusion.
>   We want to clarify that OPT is NOT the optimal value of the training problem. Instead, OPT is the smallest label prediction error of the best target function  (over the choices of parameters  $a_{\mathcal{F}, r, i}^*$, $a_{\mathcal{G}, r, i}^*$, $w^*_{r,i}$, $v_{r,i}^*$) in the concept class. OPT decreases as the concept class becomes more complex, such as increasing the integers $p_\mathcal{F}, p_\mathcal{G}$ and making activations  $\mathcal{F}$ and $\mathcal{G}$ more complex.
>
> 2. The definition of OPT does not consider learning, we only discuss the approximation accuracy of using some target function to approximate the labels. We will later use OPT to measure the generalization performance of our learned model.   We realized the original presentation sequence of defining OPT after introducing the training algorithm may lead to confusion. We adjusted the sequence to define OPT immediately after the concept class and revised the definition based on the above discussion.
>
> ## Question 2:
> In Lemma 3.1 and Theorem 3.2: It is not clear with respect to which event the probability is taken.
> ## Answer Question 2:
> Here the probability is with respect to the randomness in the SGD algorithm. The terminology is commonly used in theoretical generalization analysis, such as  Theorem 3.1 in [1], and Theorem 1 in [2].
>
> ## Question 3:
> In Theorem 3.2: Could the authors present the assumptions better or give more intuitions?
> ## Answer Question 3:
> As shown in Weakness 6, we have simplified the assumptions in  Theorem 3.2 to improve the presentation and we write a Proof overview in Appendix section C.1 to give more intuitions.
>
> ## Question 4:
> Why do the authors average in Equation 14 over all iterations of the SGD steps?
> ## Answer Question 4:
> This is a typical way to characterize the learning performance of SGD. Please see Theorem 1 in section 3.1 of paper [1] and Theorem 1 in section 6 of paper [2] also characterize the average performance of all SGD iterations. The intuition is that because the average performance is already good, considering that the models in the initial few iterations do not perform well, then the models learned at the end must have a desirable generalization.
>
> ## Question 5:
> How is $ (X, y_n) $ sampled, and how is $ D $ defined?
> ## Answer Question 5:
> Note that our results are distribution-free. The setup of $(X, y_n)$ and $\mathcal{D}$ are exactly the same as those in [3] see the first paragraph of page 7 of that paper. Specifically, in our paper, let $\mathcal{D}_{{\tilde{x}_n}}$ and
> $D_y$ denote the distribution from which the feature and label of node $n$ are drawn, respectively.
> Let $\mathcal{D}$ denote the concatenation of these distributions.
>
> Then the given feature matrix $X$ and partial labels in   $\Omega$    can be viewed as  $|\Omega|$    identically distributed but correlated samples  $(X, y_n)$ from $\mathcal{D}$. The correlation results from the fact that the label of node $i$ depends on not only the feature of node $i$ but also neighboring features. We added this discussion after equation (10) in the revision.
>
> ## Question 6:
> Could you elaborate on many assumptions, e.g., Section 3.4: it doesn't seem clear that the norms of the learned weight matrices are uniformly bounded.
> ## Answer Question 6:
> We prove that $ \|W_t\|_2 $ and $ \|V_t\|_2 $  are bounded by $\tau_w$ and $\tau_v$ in Appendix section B.3. Because $\tau_w$ and $\tau_v$ can be bounded by model complexity and sample complexity constants to simply the presentation of the paper, we removed $\tau_w$ and $\tau_v$ from the main text and replace them with their bounds using $ C_F  $ and $ C_G  $.
>
> ## Reference
> 1. Allen-Zhu, Zeyuan, Yuanzhi Li, and Yingyu Liang. "Learning and generalization in overparameterized neural networks, going beyond two layers." Advances in neural information processing systems 32 (2019).
> 2. Allen-Zhu, Zeyuan, and Yuanzhi Li. "What can resnet learn efficiently, going beyond kernels?." Advances in Neural Information Processing Systems 32 (2019).
> 3. Hongkang Li, Meng Wang, Sijia Liu, Pin-Yu Chen, and Jinjun Xiong. Generalization guarantee of training graph convolutional networks with graph topology sampling. In International Conference on Machine Learning (ICML), pp. 13014–13051. PMLR, 2022a.

---

### Official Review · Reviewer_urgn · 2023-11-10

**Soundness:** 3 good
**Presentation:** 3 good
**Contribution:** 3 good
**Rating:** 6
**Confidence:** 3

**Summary:**

The paper analyzes the generalization errors of a 2-layer graph convolutional network (GCN) that incorporates skip connections and independently performs edge sampling at two different layers. Based on the developed theorems, the paper also presents a list of practical insights. Furthermore, the paper includes experimental evaluations on synthetic datasets as well as two real-world datasets, with the experimental results aligning with the theoretical findings.

**Strengths:**

1. The analyses focus on GCN structures with skip connections, utilizing edge sampling as the sampling strategy, which distinguishes them as novel aspects compared to previous generalization analyses on GCNs.

2. Skip connections and edge sampling are two commonly adopted design elements in contemporary GCNs. The theoretical discoveries offer valuable practical insights for the development of GCN architectures.

3. The experiments are conducted on both synthetic datasets and real-world datasets, results from both experiments support the theoretical findings.

**Weaknesses:**

$\newcommand{\sB}{\mathcal{B}}$
$\newcommand{\sF}{\mathcal{F}}$
$\newcommand{\sG}{\mathcal{G}}$
$\newcommand{\mW}{\mathbf{W}}$
$\newcommand{\mV}{\mathbf{V}}$


Firstly, I want to acknowledge that I understand the challenges associated with presenting mathematically intensive theoretical analyses, and the paper's overall structure is well-constructed. The following suggestions represent some "nice-to-have" additions that could enhance the logical flow and improve reader comprehension.

1. I recommend adding an explanation for the choice of the value $d_1\sqrt{\frac{d_i}{d_j}}$ and the rationale behind differentiating the sampling strategies based on the cases where $i > j$ and $i \leq j$

2. Some conclusions are presented but not utilized within the main paper, such as the bounds on $\sB_{\sF \circ \sG}$, $||\mW_t||_2$ and $||\mV_t||_2$. This may lead to confusion regarding their initial inclusion.

3. I recommend separating the proof for Lemma 3.1 from the proof for Theorem 3.2 and integrating them within the main paper. This adjustment is essential as it contributes to one of the key insights of the paper.

**Questions:**

$\newcommand{\sL}{\mathcal{L}}$
$\newcommand{\sC}{\mathcal{C}}$
$\newcommand{\sS}{\mathcal{S}}$

1. Could the authors kindly provide a brief proof for the bound on $\sL_\sG$? I am particularly interested in the steps which introduce $\sB_{\sF}$ into the final expression.

2. The upper bound for the combination factor $\alpha$ is $O(\frac{1}{kp_\sG \sC_\sS(\sG, \sB_{\sF}||A^*||_1)})$. I am curious about the order of magnitude of this value. The concern arises when this value becomes exceedingly small in practice, which can result in the target function degrading to $\sF(A^*,x)$ and thus diminishing the potential impact of $\sG(A^*, x)$ in reducing the error. This can also lead to minimal constraints on $A^2$. However, in such cases, it deviates significantly from the concept of hierarchical learning, rendering it a trivial situation.

---

> ### Author Response · Authors · 2023-11-23
>
> ## Weakness 1:
> The analyses focus on GCN structures with skip connections, utilizing edge sampling as the sampling strategy, which distinguishes them as novel aspects compared to previous generalization analyses on GCNs.
> ## Answer Weakness 1:
> Our core sampling concept revolves around sampling edges between nodes of lower degrees with higher probability. Because the edges between lower-degree nodes correspond to larger entries in $A$, we want to sample larger entries with a higher probability and smaller entries with a lower probability. In our sampling strategy, we just divide entries in $A$ into two parts, the parts with larger values are sampled with $1-p_{ij}^k$, and the group with smaller values is sampled with $p_{ij}^k$, where $p_{ij}^k$. The size of the larger part, which is selected as $d_1\sqrt{d_i/d_j}$, is selected to ensure that $ \left \| A^* \right \| _1 $ is $O(1)$, which is required in our generalization analysis. It does not have to the exact value of $d_1\sqrt{d_i/d_j}$, any value in the order of $d_1\sqrt{d_i/d_j}$ will not change our order-wise analysis.
>
> We want to emphasize that although we set these value to simplify our theoretical analysis, the main idea of
> sampling edges between nodes of lower degrees with higher probability are preserved in our sampling strategy. We added a footnote about this in the paper.
>
> ## Weakness 2:
> Some conclusions are presented but not utilized within the main paper. This may lead to confusion regarding their initial inclusion.
> ## Answer Weakness 2:
> We have removed them in the main paper to simplify presentation.
>
> ## Weakness 3:
> I recommend separating the proof for Lemma 3.1 from the proof for Theorem 3.2 and integrating them within the main paper. This adjustment is essential as it contributes to one of the key insights of the paper.
> ## Answer Weakness 3:
> We separated the proof for Lemma 3.1 We also added a proof overview of Theorem 3.1 and Lemma 3.1. Due to the space limit of the main text, we put it in Appendix Section B.1.
>
> ## Question 1:
> Could the authors kindly provide a brief proof for the bound on $ \mathcal{L}_G $? I am particularly interested in the steps which introduce $ \mathcal{B}_F $ into the final expression.
>
> ## Answer Question 1:
> We add the proof in Appendix Section A.
>
> ## Question 2:
> The upper bound for the combination factor $\alpha$ is $O\left(\frac{1}{k_{pg}C_s(\mathcal{G},\mathcal{F}\|A^*\|_1)}\right)$. I am curious about the order of magnitude of this value. The concern arises when this value becomes exceedingly small in practice, which can result in the target function degrading to $\mathcal{F}(A^*, x)$ and thus diminishing the potential impact of $\mathcal{G}(A^*, x)$ in reducing the error. This can also lead to minimal constraints on $A^2$. However, in such cases, it deviates significantly from the concept of hierarchical learning, rendering it a trivial situation.
>
>
> ## Answer Question 2:
> We completely understand the reviewer's concern that our analysis only specifies the order but not the magnitude, and  a small magnitude of $\alpha$ can make the second term much smaller than the first term in the target function. } We agree that the result would be stronger if we could specify the magnitude of $\alpha$ and prove it to be large. However, we need to emphasize that this is almost impossible because of the high complexity of the analysis in this paper, and we highly doubt if it is possible at all. In fact, the generalization analysis for a two-layer ResNet in [1] also only specifies the order of $\alpha$ (See Theorem 1 in Section 6). Moreover, even if $\alpha$ is small such that the base function $\mathcal{F}$ can predict the labels with reasonable accuracy, adding the additional component $\alpha \mathcal{G}(\mathcal{F})$ can still improve the accuracy, and our result provides the theoretical guarantee of the hierarchical learn these two functions.
>
> Furthermore, in our experiments in Section 4.1, we choose $\alpha=0.5$. This value is deliberately chosen to be non-negligible to preserve the hierarchical structure that is central to our model's learning process. A significant $\alpha$ ensures that the contribution of the function $\| \mathcal{F} \|_2$ is on the same order of magnitude as $\| \alpha \mathcal{G}(\mathcal{F}) \|_2)$.
>
> ## Reference
> 1. Allen-Zhu, Zeyuan, and Yuanzhi Li. "What can resnet learn efficiently, going beyond kernels?." Advances in Neural Information Processing Systems 32 (2019).

---

### Meta-Review · Area_Chair_kDwv · 2023-12-06

**Metareview:**

The paper analyzes the generalization errors of a 2-layer GCN with skip connections and edge sampling as its design elements. The authors propose a sampling strategy that varies between layers and provides theoretical results for the performance of GCNs trained with SGD. Experimental evaluations on synthetic and real-world datasets align with the theoretical findings.

Strengths of the paper:
* The paper addresses an interesting and important problem by analyzing the impact of skip connections and edge sampling on the generalization performance of GCNs, distinguishing it from previous generalization analyses.
* The paper provides solid theoretical results, which is a strength in understanding the behavior of GCNs with skip connections and edge sampling.

Weaknesses of the paper:
* Reviewers noted that the paper tends to overstate its results, particularly in claiming to analyze a large class of graph learning models when it focuses mainly on two-layer GCNs.
* Some of the assumptions made in the paper are considered highly restrictive, such as assuming perfectly homophilous graphs, which may not reflect real-world scenarios.
* Reviewers found issues with clarity, missing notations and definitions, and insufficient explanations in various parts of the paper, which can hinder understanding. The presentation of the paper is criticized for introducing numerous notations without proper explanation, making it difficult to follow.
* The paper's focus on dense graphs and two-layer skip connections may limit its practical relevance, as real-world graphs are often sparse, and more complex architectures are used in practice.

**Justification For Why Not Higher Score:**

This review further highlights the paper's weaknesses in terms of presentation, clarity, and understanding of prior work, and it suggests that improvements are needed to address these issues.

**Justification For Why Not Lower Score:**

N/A.

---

### Decision · Program_Chairs · 2024-01-16

Reject